# Multivesicular bodies mediate long-range retrograde NGF-TrkA signaling

**Mengchen Ye[1,2], Kathryn M Lehigh[2,3], David D Ginty[2]***

[1]Human Genetics Training Program, The Johns Hopkins University, School of Medicine, Baltimore, United States; [2]Department of Neurobiology, Howard Hughes Medical Institute, Harvard Medical School, Boston, United States; [3]Department of Neuroscience, The Johns Hopkins University, School of Medicine, Baltimore, United States

**Abstract** The development of neurons in the peripheral nervous system is dependent on target-derived, long-range retrograde neurotrophic factor signals. The prevailing view is that target-derived nerve growth factor (NGF), the prototypical neurotrophin, and its receptor TrkA are carried retrogradely by early endosomes, which serve as TrkA signaling platforms in cell bodies. Here, we report that the majority of retrograde TrkA signaling endosomes in mouse sympathetic neurons are ultrastructurally and molecularly defined multivesicular bodies (MVBs). In contrast to MVBs that carry non-TrkA cargoes from distal axons to cell bodies, retrogradely transported TrkA[+] MVBs that arrive in cell bodies evade lysosomal fusion and instead evolve into TrkA[+] single-membrane vesicles that are signaling competent. Moreover, TrkA kinase activity associated with retrogradely transported TrkA[+] MVBs determines TrkA[+] endosome evolution and fate. Thus, MVBs deliver long-range retrograde NGF signals and serve as signaling and sorting platforms in the cell soma, and MVB cargoes dictate their vesicular fate.

DOI: https://doi.org/10.7554/eLife.33012.001

*For correspondence:
david_ginty@hms.harvard.edu

## Introduction

Developing neurons in the peripheral nervous system (PNS) are critically dependent on target-derived trophic cues released by the end organs they innervate. This is best exemplified for primary somatosensory neurons and postganglionic sympathetic neurons and their dependence on members of the neurotrophin family of growth factors, which includes nerve growth factor (NGF), brain-derived neurotrophic factor (BDNF), neurotrophin-3 (NT3) and neurotrophin-4 (NT4). Through binding to and activating their cognate receptors (NGF to TrkA, BDNF and NT4 to TrkB, NT3 predominantly to TrkC), neurotrophins promote neuronal survival, axonal growth and target field innervation, neuronal subtype specification and, for sympathetic neurons, synapse formation and maintenance (*Cosker et al., 2008*; *Harrington and Ginty, 2013*; *Huang and Reichardt, 2001*; *Luo et al., 2007*; *Miller and Kaplan, 2001*; *Sharma et al., 2010*).

NGF is the first described and prototypical neurotrophin. A remarkable feature of NGF signaling is that, in order to elicit trophic actions on developing sensory and sympathetic neurons, the NGF/TrkA signal must be propagated retrogradely, from distal axons to the soma, traveling a distance that is thousands of times the diameter of the cell body (*Cosker et al., 2008; Howe and Mobley, 2005*). Although much is known about downstream signaling pathways and transcriptional targets that mediate the trophic actions of NGF, the molecular events that orchestrate this unique form of long-range axonal signaling remain unclear and debated.

A major mode of retrograde NGF signal propagation is through the actions of NGF/TrkA signaling endosomes. The signaling endosome model posits that NGF/TrkA complexes formed at distal axons are internalized and actively sorted into endosomes, a subset of which become mature,

signaling-competent vesicles that undergo long-distance, microtubule-mediated retrograde transport to the soma (*Howe and Mobley, 2005*; *Ye et al., 2003*; *Heerssen et al., 2004*; *Wu et al., 2007*). Indeed, experiments in which TrkA receptors were labeled in distal axons prior to internalization revealed their subsequent appearance in proximal axons, cell bodies and dendrites of sympathetic neurons in vitro (*Sharma et al., 2010*; *Harrington et al., 2011*). By recruiting TrkA effectors, TrkA endosomes in these different cellular compartments can function as signaling platforms that propagate trophic signals to support survival, maturation, and synapse formation and maintenance (*Heerssen et al., 2004*; *Harrington et al., 2011*; *Howe et al., 2001*; *Suo et al., 2014*).

Defining molecular and ultrastructural features of signaling endosomes is key to understanding the mechanisms that underlie long-range NGF trafficking and signaling. A long-standing, canonical view is that retrograde NGF/TrkA signaling endosomes are early endosomes because, like other ligand/receptor complexes, internalized NGF/TrkA complexes are sorted into early endosomes and the membrane topology of receptors in early endosome-type vesicles can, in principle, enable signal propagation within the cytoplasm (*Cosker et al., 2008*; *Harrington and Ginty, 2013*; *Howe and Mobley, 2005*). Evidence supporting the early endosome model includes biochemical and ultrastructural analyses using an ex vivo sciatic nerve preparation (*Delcroix et al., 2003*). The small GTPase Rab5, a molecular marker and master regulator of early endosomes (*Zerial and McBride, 2001*), is enriched in purified axoplasmic fractions that contained retrograde $^{125}$I-NGF. Also, TrkA was found associated with single-membrane vesicles labeled by Rab5 (*Delcroix et al., 2003*). In agreement with this, retrogradely transported NGF conjugated to quantum dots was colocalized with Rab5 in compartmentalized cultures of dorsal root ganglion (DRG) neurons (*Cui et al., 2007*). However, despite the widely accepted view that signaling endosomes are early endosomes, late endosomes (LEs) and multi-vesicular bodies (MVBs) have been observed to contain retrogradely transported cargoes in other experimental paradigms. For example, in cultured DRG and motor neurons, internalized tetanus toxin is sorted into TrkB$^+$ vesicles associated with either Rab5 or Rab7 (a small GTPase enriched in MVBs and LEs) within axons, but only Rab7$^+$ vesicles underwent long-range retrograde transport (*Deinhardt et al., 2006*). In addition, in compartmentalized sympathetic neuron cultures, $^{125}$I-NGF applied to distal axons was found predominantly in MVBs and lysosomes, based on ultrastructural analysis (*Claude et al., 1982a*, *1982b*). In related work, gold-labeled NGF injected into the anterior chamber of the eye accumulated in MVBs in axons near sympathetic ganglia (*Sandow et al., 2000*). Also, activated TrkA was observed in multivesicular structures in the rat sciatic nerve (*Bhattacharyya et al., 2002*). However, BDNF injected into the tongue did not appear associated with MVBs or any other ultrastructurally defined endosome type in hypoglossal motor neurons (*Altick et al., 2009*). Moreover, a key conceptual challenge to the idea that MVBs are TrkA signal carriers is the lack of a plausible mechanism to explain how signals emanating from NGF/TrkA complexes encapsulated within intraluminal vesicles of MVBs are transmitted to the cytoplasm. Thus, while different endosomal compartments have been suggested as candidate retrograde NGF transport carriers, possibly due to the different means used to visualize endosomes, different preparations, and the use of different cell types, the prevailing view is that early endosomes are retrograde TrkA$^+$ signaling endosomes.

Here, we demonstrate that molecularly and ultrastructurally defined MVBs carry the majority of NGF/TrkA complexes from distal axons to cell bodies. Moreover, Rab7, which regulates MVB and LE function (*Zerial and McBride, 2001*; *Vanlandingham and Ceresa, 2009*; *Lebrand et al., 2002*), is essential in vivo and in vitro for retrograde TrkA signaling. Strikingly, TrkA$^+$ MVBs that arrive in cell bodies do not fuse with lysosomes, but rather can give rise to single-membrane vesicles marked by Vps35, but not Rab5. Moreover, TrkA kinase activity controls the maturation and fate of TrkA$^+$ MVBs. Thus, MVBs are mediators of long-range retrograde TrkA signaling, evading lysosomal fusion and degradation within the soma in a TrkA kinase-dependent manner, and evolving within the soma into single membrane vesicles with the capacity to propagate TrkA survival signals.

## Results

### Ultrastructural analysis of retrograde Flag-TrkA endosomes

We investigated the vesicular nature, molecular composition, maturation, and signaling dynamics of TrkA$^+$ signaling endosomes using a *Ntrk1$^{Flag}$* knockin mouse line, which expresses Flag epitope-

tagged TrkA from the endogenous TrkA locus, and an in vitro compartmentalized microfluidic sympathetic neuron culture system to monitor internalization, sorting and retrograde trafficking of Flag-TrkA$^+$ endosomes (*Figure 1—figure supplement 1A*) (*Sharma et al., 2010*; *Harrington et al., 2011*). We first sought to define the ultrastructural features of retrogradely transported TrkA$^+$ endosomes. To accomplish this, compartmentalized *Ntrk1$^{Flag}$* sympathetic neurons and an anti-Flag antibody pre-conjugated to Protein A-5nm gold were used to visualize retrogradely transported Flag-TrkA$^+$ endosomes by electron microscopy (EM). While application of neither the primary antibody nor Protein A-5nm gold alone to distal axons of compartmentalized *Ntrk1$^{Flag}$* neurons yielded electron-dense structures detectable by EM, application of anti-Flag antibody that was pre-conjugated to Protein A-5nm gold to *Ntrk1$^{Flag}$* neurons, but not wild-type neurons, labeled electron dense structures in axons that were readily apparent by EM (*Figure 1A*). This antibody labeling strategy did not perturb normal internalization and endocytic trafficking of TrkA receptors; Gold-labeled TrkA receptors did not compromise survival or retrograde signaling (data not shown), nor did it affect the subcellular and ultrastructural localization of P-TrkA, visualized by either light microscopy or EM (see below, Figures 4 and 6).

As expected, newly internalized, gold-labeled TrkA receptors in distal axons were found in single-membrane vesicular structures in close proximity to the plasma membrane, which is a defining feature of early endosomes (*Figure 1—figure supplement 1B*). Surprisingly, and in stark contrast, retrogradely transported TrkA receptors in proximal axons and cell bodies were found mainly in MVBs (87.9 ± 5.0%) and, to a much lesser extent, single-membrane vesicles (SVs, 9.5 ± 2.8%) or lysosomes (2.6 ± 1.3%; *Figure 1B*, *Figure 1—figure supplement 1B,C*). The gold-labeled Flag-TrkA receptors were localized both to the limiting membrane and intraluminal vesicles (ILVs) of MVBs (*Figure 1B*). Therefore, following NGF treatment of distal axons, newly internalized TrkA in distal axons is associated with early endosomes, whereas following retrograde transport to proximal axons and cell bodies TrkA is predominantly associated with MVBs.

## Internalized TrkA is sorted into both early endosomes and MVBs in distal axons, and the majority of retrogradely transported TrkA$^+$ endosomes are MVBs

To complement the EM analyses and to define molecular features of TrkA signaling endosomes in sympathetic neuron axons, we next used the Flag-TrkA transport assay to visualize TrkA$^+$ endosomes and determine the extent to which they are associated with markers of distinct types of vesicular compartments. Consistent with the EM observations, following application of NGF exclusively to distal axons, newly internalized TrkA receptors in distal axons were mainly associated with Rab5-labeled early endosomes (*Figure 2A*). However, the fraction of total internalized Flag-TrkA colocalized with Rab5 in distal axons decreased during the next hour, coincident with an increase of Flag-TrkA colocalized with Rab7, a small GTPase that regulates the functions of MVBs and late endosomes (LEs) (*Figure 2—figure supplement 1A*) (*Vanlandingham and Ceresa, 2009*; *Lebrand et al., 2002*). Strikingly, Flag-TrkA punctae within proximal axons, which represent mature, retrogradely transported endosomes, were mainly associated with MVB/LE markers, including Rab7, Hrs (*Henne et al., 2011*) and CD63 (*Pols and Klumperman, 2009*) and, to a much lesser extent, with markers of early endosomes (Rab5), recycling endosomes (Rab11) and lysosomes (Lamp1) (*Figure 2A,B*; *Figure 2—figure supplement 1B,C*).

We next directly visualized axonal transport of TrkA endosomes in real time by imaging *Ntrk1$^{Flag}$* neurons expressing fluorophore-tagged endosomal proteins using a modified Flag-TrkA assay (*Figure 2C–E*). In this live cell imaging paradigm, anti-Flag antibody pre-conjugated with fluorescent secondary antibody was applied to distal axons of *Ntrk1$^{Flag}$* neurons. This live cell Flag-TrkA imaging assay enabled specific visualization of Flag-TrkA endosomes because fluorescent signals were undetectable in axons of wild-type neurons (data not shown). We expressed in *TrkA$^{Flag}$* neurons EGFP-tagged Rab5 or Rab7 using a lentiviral delivery system. These constructs did not compromise NGF-dependent survival of sympathetic neurons (*Figure 2—figure supplement 1D*). Live cell imaging of EGFP-Rab5-expressing *Ntrk1$^{Flag}$* neurons revealed that the majority of EGFP-Rab5$^+$/Flag-TrkA endosomes in distal axons are stationary or move bi-directionally for only a short distance (*Figure 2C*, yellow arrow heads). Consistent with this, only few of the EGFP-Rab5$^+$/Flag-TrkA$^+$ punctae were co-transported retrogradely into proximal axons of the cell body compartment (*Figure 2C–E*; *Video 1*). In contrast, the majority of retrogradely transported Flag-TrkA$^+$ endosomes observed within

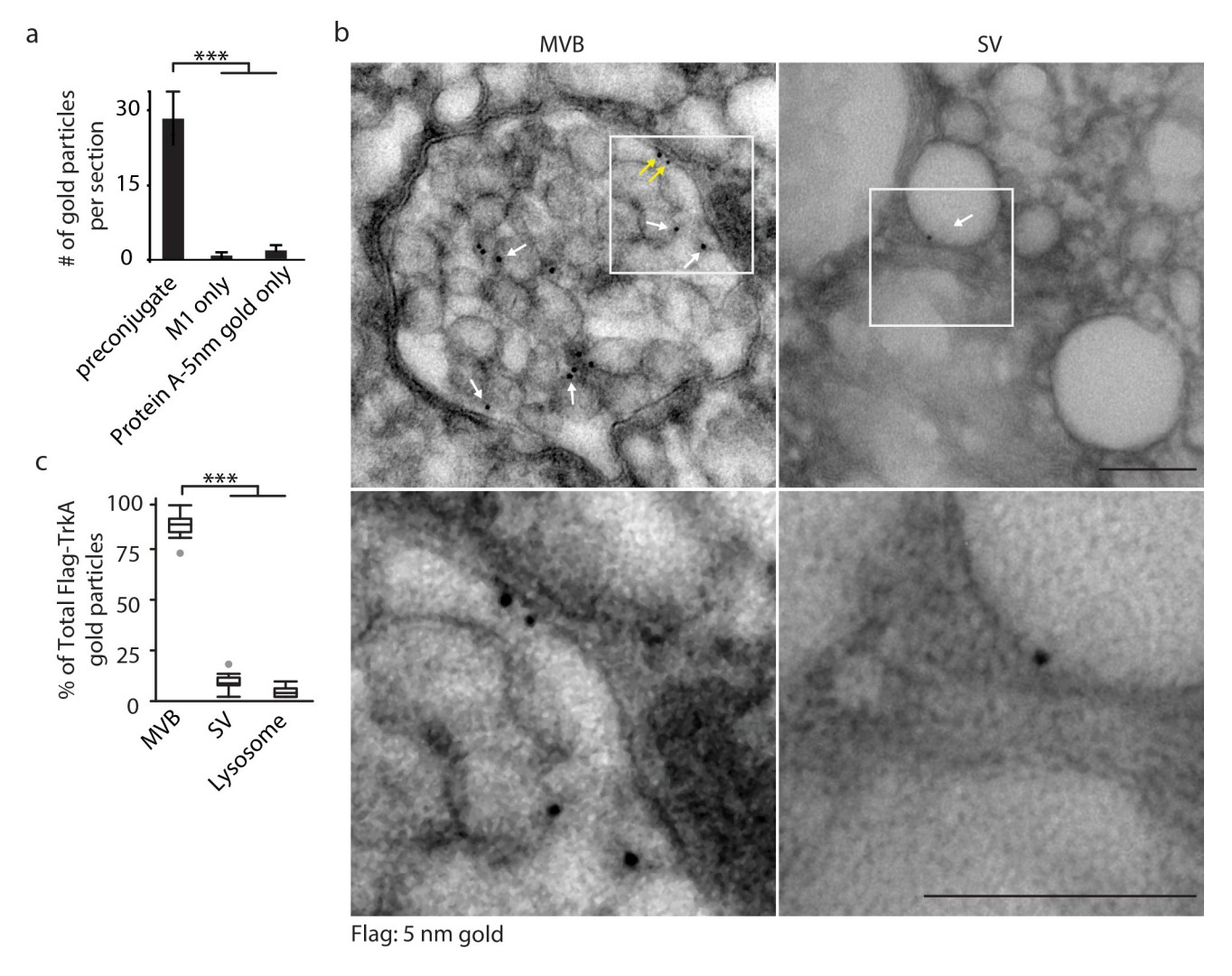

Flag: 5 nm gold

**Figure 1.** Retrograde TrkA[+] endosomes are predominantly of multi-vesicular, not single-vesicular, ultrastructure. (a,b) The Flag-TrkA transport assay was performed in compartmentalized sympathetic neurons using pre-conjugated anti-Flag antibody with Protein A-5 nm gold. Cells were fixed 1 hr post-NGF application and processed for EM. The percentage of Flag-TrkA gold particles localized to MVBs, single-membrane vesicles (SVs) or lysosomes was quantified (c). Note the presence of the Flag epitope on both the membrane of the intraluminal vesicles (white arrows) and the limiting membrane of the MVB (yellow arrows). High-magnification images of the boxed areas are shown in the bottom panels. (c) The Flag-TrkA assay was performed using pre-conjugated primary antibody and Protein A-5nm gold, or primary antibody or Protein A-5nm only. Cells were fixed 1 hr post-NGF stimulation and the number of gold particles per EM section was counted (n = 4). Scale bar: 100 nm. Data are represented as mean ± standard error of the mean (SEM) (a) or presented in box plot (c). In box plots, the top and the bottom of the central rectangle represents the 75th and 25th percentile value, respectively, and the line inside represents the median; the whisker on either side extends to the data point that is within the range of variation (1.5×(75th percentile – 25th percentile)) and data points beyond that range are plotted as individual dots. ***p<0.001 by one-way ANOVA with a Tukey's *post-hoc* test. See also *Figure 1—figure supplement 1*.

DOI: https://doi.org/10.7554/eLife.33012.002

The following figure supplement is available for figure 1:

**Figure supplement 1.** Retrogradely transported TrkA is associated with MVBs.

DOI: https://doi.org/10.7554/eLife.33012.003

proximal axons were associated with EGFP-Rab7 (*Figure 2C*, white arrow heads). Furthermore, unlike EGFP-Rab5[+]/Flag-TrkA[+] endosomes, the majority of EGFP-Rab7[+] TrkA endosomes exhibited processive movement in the retrograde direction; very few were stationary or moved bi-directionally (*Figure 2C–E*; *Video 2*). The percentages of Flag-TrkA punctae co-transported with either EGFP-Rab5 or EGFP-Rab7 were similar to the percentages of colocalization observed in the fixed cell

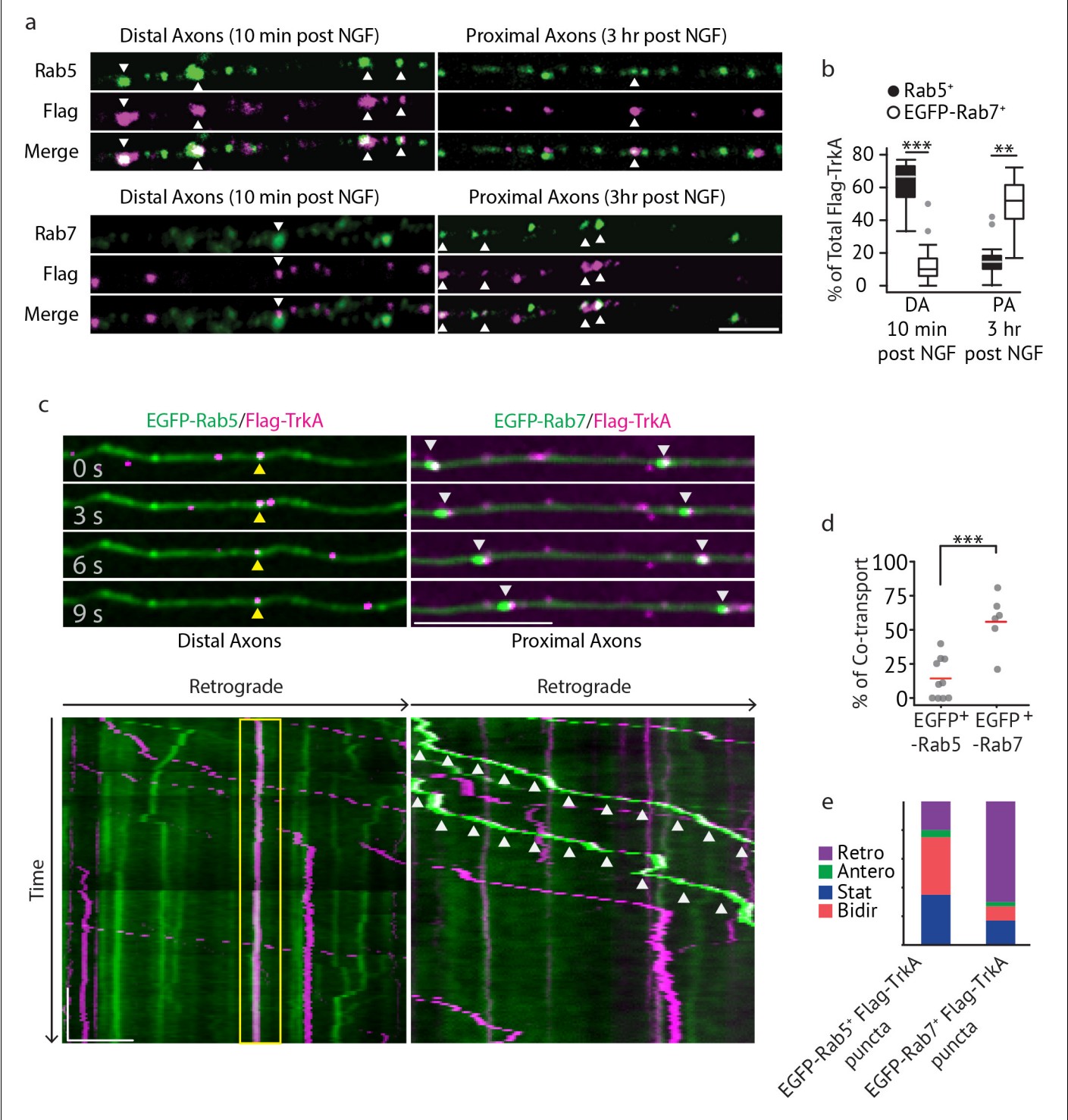

**Figure 2.** Multivesicular bodies, not early endosomes, are major carriers of retrograde TrkA signals in sympathetic neurons. (a,b) The Flag-TrkA endosome transport assay was performed in sympathetic neurons grown in compartmentalized microfluidic culture and infected with a lentivirus expressing EGFP-Rab7. The percentage of Flag-TrkA punctae (magenta) colocalized with the MVB marker EGFP-Rab7 or the EE marker Rab5 (green) in distal axons (DA) 10 min post NGF application or in axons proximal to cell bodies (PA) 3 hr post NGF application was quantified (n = 5). For DA experiments, cells were washed with NaCl/acetic acid to remove surface bound Flag signal prior to fixation. Scale bar: 5 μm. (c–e) Sympathetic neurons grown in compartmentalized culture were infected with lentivirus expressing EGFP-Rab7 or EGFP-Rab5. The Flag-TrkA assay was performed using pre-conjugated anti-Flag antibody and Alexa Fluoro secondary antibody during the 4°C incubation step. Flag-TrkA (magenta) and EGFP (green) were then imaged consecutively in axons. Representative time-lapse images of each type of TrkA endosomes are shown (top panel). Kymographs of time-lapse

*Figure 2 continued on next page*

*Figure 2 continued*

images are shown in the bottom panel. Arrowheads denote individual endosomes. Scale bar: 10 µm; 1 min. The percentage of Flag-TrkA punctae co-transported with either marker was quantified (d). Note that not every axon expresses the endosomal marker fusion proteins and therefore Flag-TrkA in these axons was not always observed to be associated with fluorophore-tagged endosomal markers. The directionality of retrograde TrkA MVBs and EEs are shown in (e). A total of 168 (Rab5) and 207 (Rab7) endosomes were scored in four independent experiments for each condition. Data are presented in box plot (b) or dot plot (d). In dot plots, individual data points (dots) and the mean (red line) are shown. ***p<0.001 by two tailed unpaired Student's *t* test. See also *Figure 2—figure supplement 1* and *Videos 1* and *2*.

DOI: https://doi.org/10.7554/eLife.33012.004

The following figure supplement is available for figure 2:

**Figure supplement 1.** Retrogradely transported TrkA is associated with MVB markers.

DOI: https://doi.org/10.7554/eLife.33012.005

immunolabeling experiments. Similar results were obtained using a different MVB marker, CD63 (*Figure 2—figure supplement 1E*; *Video 3*). These observations, taken together, demonstrate that the majority of retrograde Flag-TrkA$^+$ endosomes are associated with markers of MVBs or LEs, and not early endosomes.

## Rab7 mediates retrograde TrkA transport and signaling in vivo and in vitro

We previously found that Rab7, a small GTPase that regulates functions of MVBs and LEs (*Vanlandingham and Ceresa, 2009*; *Lebrand et al., 2002*), is associated with immunopurified TrkA$^+$ endosomes (*Harrington et al., 2011*). Moreover, Rab7 is a major regulator of vesicular trafficking (*Wandinger-Ness and Zerial, 2014*). In motor neurons, retrograde transport of a modified tetanus toxin is mediated by Rab7$^+$ endosomes and is dependent on Rab7's GTPase activity (*Deinhardt et al., 2006*). Therefore, we asked whether Rab7 mediates long-range retrograde TrkA trafficking and signaling in sympathetic neurons using a combination of in vivo and in vitro analyses. Mice harboring a $Th^{2a-CreER}$ knock-in allele (*Abraira et al., 2017*) and a conditional allele of *Rab7* (*Roy et al., 2013*) were generated to examine the role of Rab7 in sympathetic neurons in vivo (*Figure 3A–D*). To ask whether Rab7 plays a role in NGF-dependent survival signaling in sympathetic neurons during early stages of development, $Th^{2a-CreER}$; $Rab7^{f/f}$ mice were treated with tamoxifen at E14, resulting in a partial loss of Rab7 protein during the critical period of NGF-dependent sympathetic neuron survival (*Figure 3—figure supplement 1A*). Superior cervical ganglia (SCGs) were harvested at P7 and the number of sympathetic neurons was counted. Compared to $Th^{2a-CreER}$; $Rab7^{+/+}$ littermate controls, $Th^{2a-CreER}$; $Rab7^{f/f}$ animals exhibited ~60% neuronal cell loss (*Figure 3A,B*). To address the role of Rab7 in TrkA signaling in sympathetic neuronal soma and dendrites postnatally, $Th^{2a-CreER}$; $Rab7^{f/f}$ mice were treated with tamoxifen at P7 and P8 and SCGs were harvested at P14, a time following the period of NGF-dependent survival of sympathetic neurons. TrkA signaling in sympathetic neurons was assessed using antibodies that specifically recognize the phosphorylated TrkA tyrosine residues 490 or 785; these phosphorylation events are indicative of the active form of TrkA. Interestingly, a marked reduction of P-TrkA punctae was observed in sympathetic ganglia of $Th^{2a-CreER}$; $Rab7^{f/f}$ mice compared to $Th^{2a-CreER}$; $Rab7^{f/+}$ controls (*Figure 3C*). Moreover, in the absence of Rab7, a decrease in both pre- and post-synaptic specializations visualized using antibodies against the vesicular acetylcholine transporter (VAChT) and Homer1, respectively, was observed (*Figure 3C,D*). Therefore, Rab7 functions in sympathetic neurons in vivo to support NGF-dependent survival and synapse formation.

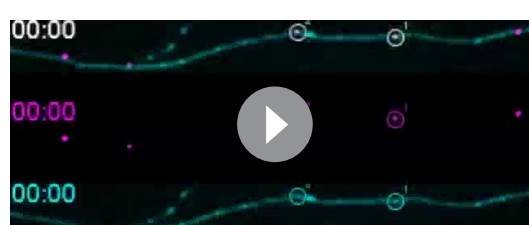

**Video 1.** EGFP-Rab5$^+$ TrkA$^+$ endosomes in distal axons are stationary/oscillatory. Time lapses of EGFP-Rab5 (cyan, bottom panel) and Flag-TrkA (magenta, middle panel) trafficking (merged channel, top panel) in distal axons of sympathetic neurons grown in compartmentalized cultures. Notice a stationary EGFP-Rab5$^+$ Flag-TrkA endosome (#1) and three retrogradely transported EGFP-Rab5$^-$ Flag-TrkA endosomes (#2–4). Retrograde is to the right. The video was acquired at 2 fps (frames per second) and played at 15 fps.

DOI: https://doi.org/10.7554/eLife.33012.006

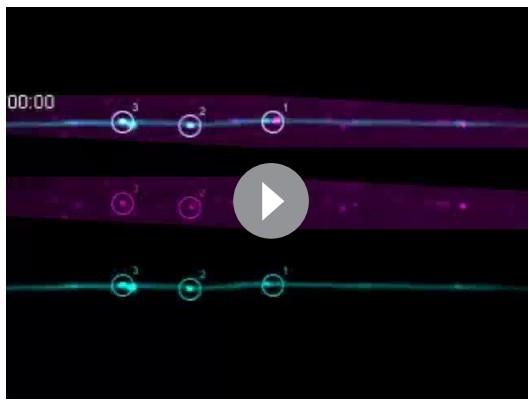

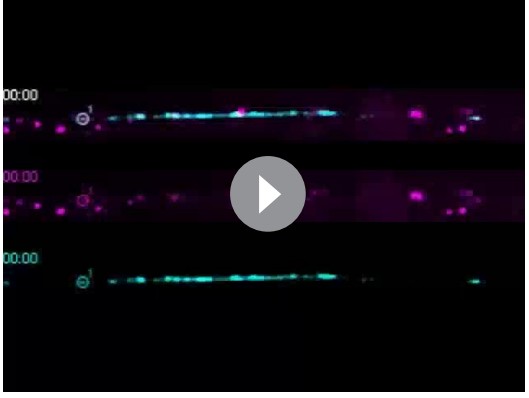

**Video 2.** EGFP-Rab7$^+$ TrkA$^+$ endosomes display consistent and processive retrograde movement in axons. Time lapses of EGFP-Rab7 (cyan, bottom) and Flag-TrkA (magenta, middle) trafficking (merged channel, top) in axons in the middle grooves of microfluidic chambers adjacent to the distal axon compartment. Shown are three retrogradely transported EGFP-Rab7$^+$ Flag-TrkA endosome (#1–3). Retrograde is to the right. The video was acquired at 2 fps (frames per second) and played at 15 fps.
DOI: https://doi.org/10.7554/eLife.33012.007

**Video 3.** CD63-EGFP$^+$ TrkA$^+$ endosomes display consistent and processive retrograde movement in axons. Time lapses of CD63-EGFP (cyan, bottom) and Flag-TrkA (magenta, middle) trafficking in axons (merged channel, top) in the middle grooves of microfluidic chambers adjacent to the distal axon compartment. Shown are two retrogradely transported CD63-EGFP$^+$ Flag-TrkA endosome (#1–2). Retrograde is to the right. The video was acquired at two fps (frames per second) and played at 15 fps.
DOI: https://doi.org/10.7554/eLife.33012.008

We next investigated the role of Rab7 in retrograde survival signaling in sympathetic neurons under in vitro conditions in which the time of application of NGF exclusively to distal axons is controlled. Rab7 was ablated in *Rab7$^{f/f}$* sympathetic neurons grown in compartmentalized chambers by infecting them with a lentivirus expressing Cre recombinase, or in wild-type compartmentalized sympathetic neurons infected with a lentivirus expressing an shRNA against Rab7, which greatly reduced Rab7 protein levels (*Figure 3—figure supplement 1B*). NGF was then applied exclusively to distal axons to evaluate the role of Rab7 in retrograde TrkA signaling. Consistent with the in vivo analysis, retrograde TrkA signaling (*Figure 3E,F*) and survival (*Figure 3G*; *Figure 3—figure supplement 1D*) were compromised in neurons lacking Rab7 in vitro. This was not due to a non-specific or global effect on neuronal survival, since addition of NGF directly to cell bodies and proximal axons of neurons lacking Rab7 supported their survival (*Figure 3G*; *Figure 3—figure supplement 1D*). Also, Rab7 knockdown did not affect mitochondria movement in axons (*Figure 3—figure supplement 1G*). To ask whether Rab7 is necessary for retrograde TrkA transport, we performed the Flag-TrkA endosome transport assay in sympathetic neurons expressing Rab7 shRNA. Using this assay, we found that expression of the Rab7 shRNA in sympathetic neurons virtually abolished retrograde transport of Flag-TrkA$^+$ endosomes (*Figure 3H,I*). On the other hand, Rab7 depletion did not affect TrkA internalization in distal axons (*Figure 3H*, DA). To assess the specificity of the Rab7 shRNA treatment, we used two Rab7 replacement constructs, one encoded by a human Rab7 cDNA that is resistant to the shRNA and another encoded by a mouse Rab7 cDNA that is the target of the shRNA. Expression of hRab7, but not mRab7, in sympathetic neurons treated with the Rab7 shRNA restored both retrograde TrkA transport and survival (*Figure 3—figure supplement 1C,D*). Similar observations were found using TrkA$^+$ DRG sensory neurons, suggesting that Rab7 is a general mediator of retrograde TrkA transport and survival across different neuronal types (*Figure 3—figure supplement 1E,F*). Taken together, these findings indicate that Rab7 mediates retrograde TrkA transport and signaling both in vivo and in vitro.

## Recruitment of RILP to TrkA MVBs mediated by Rab7 is required for retrograde TrkA transport

Rab7 has been implicated in the early-to-late endosome transition as well as the function and maturation of late endosomes (*Vanlandingham and Ceresa, 2009*; *Lebrand et al., 2002*; *Rink et al.,*

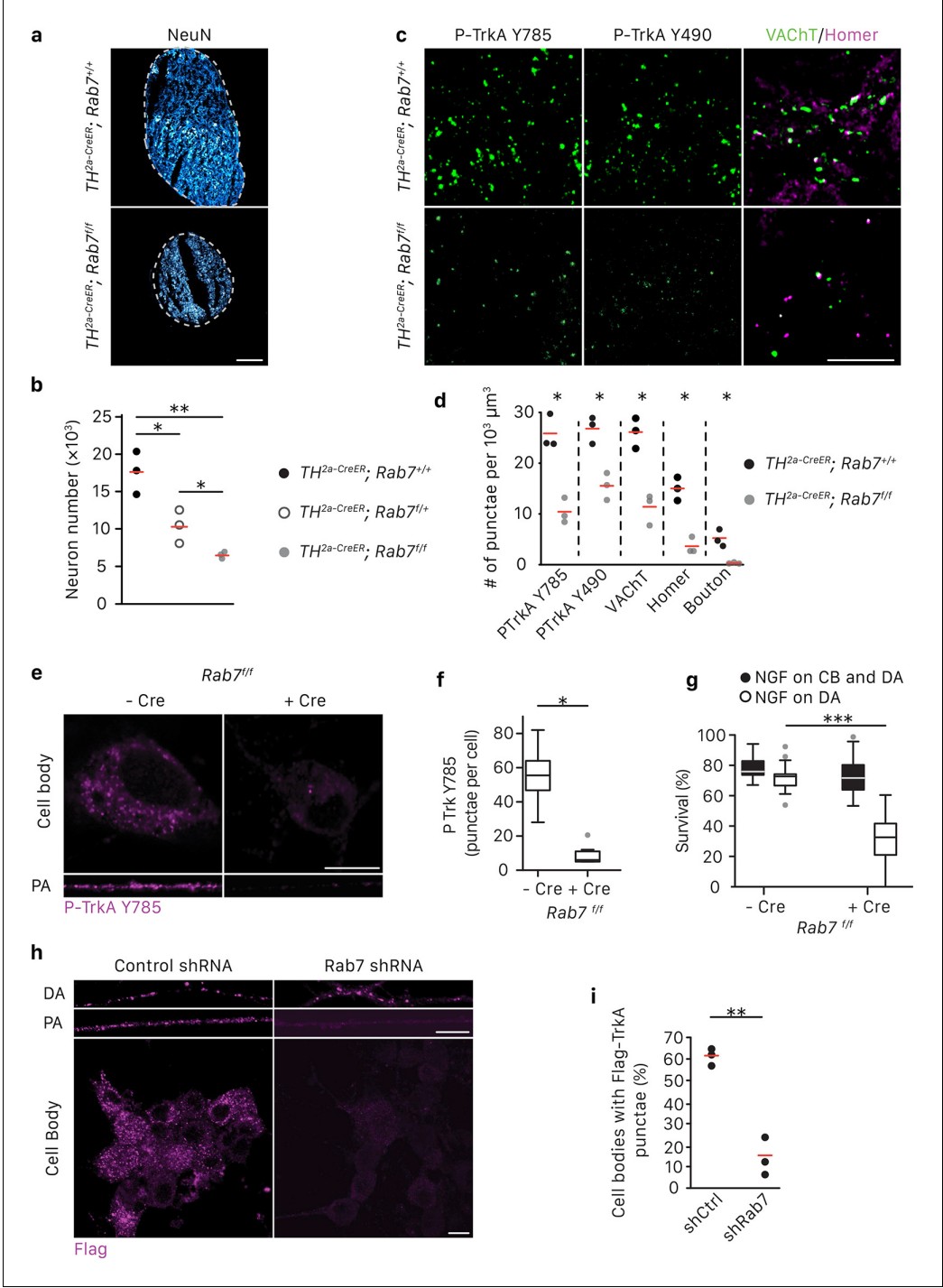

**Figure 3.** Rab7 mediates survival and synaptogenesis of sympathetic ganglia in vivo and retrograde NGF/TrkA transport, signaling and survival in vitro. (a,b) Neuronal cell counts of SCGs from $Rab7^{+/+}$; $Th^{2a-CreER}$, $Rab7^{f/+}$; $Th^{2a-CreER}$ and $Rab7^{f/f}$; $Th^{2a-CreER}$ mice at P7. Tamoxifen was administered at E14 (0.5 mg) to induce Cre expression (n = 3). Scale bar: 100 μm. (c,d) $Rab7^{+/+}$; $Th^{2a-CreER}$ and $Rab7^{f/f}$; $Th^{2a-CreER}$ mice were treated with 1 mg tamoxifen at P7 and SCGs were harvested at P14. TrkA signaling was assessed by P-Trk Y490 and Y785 staining. Synaptic organization was assessed by VAChT (green) and Homer (magenta) staining, and VAChT/Homer colocalization (n = 3). (e–g) Sympathetic neurons harvested from P0 $Rab7^{f/f}$ pups were grown in microfluidic chambers and infected with a lentivirus expressing the Cre recombinase. Cells were incubated with NGF in the distal axon compartment and anti-NGF and the caspase inhibitor, BAF, in the cell body compartment for 48 hr, and TrkA signaling was assessed in axons and cell soma was assessed by P-Trk Y785 immunostaining (e,f). Alternatively,

*Figure 3 continued on next page*

*Figure 3 continued*
retrograde NGF-dependent neuronal survival was assessed (g) (n = 3). (h,i) The Flag-TrkA transport assay was performed in sympathetic neurons infected with lentivirus expressing either a control shRNA or an shRNA against Rab7. The accumulation of Flag-TrkA punctae in cell bodies, which represent retrogradely transported TrkA, was assessed (n = 3). Scale bar: 10 µm (c,e,h). Data are presented in dot plots (b,d,i) or box plots (f,g). *p<0.05, **p<0.01 and ***p<0.001 by one-way ANOVA with a Tukey's *post-hoc* test (b,g) or a two tailed unpaired Student's *t* test (d,f,i). See also *Figure 3—figure supplement 1*.
DOI: https://doi.org/10.7554/eLife.33012.009
The following figure supplement is available for figure 3:

**Figure supplement 1.** Rab7 is required for retrograde TrkA transport and survival in sympathetic and sensory neurons.
DOI: https://doi.org/10.7554/eLife.33012.010

*2005*). To define the step at which Rab7 functions in the context of TrkA endosome maturation, we tested the necessity of Rab7 for formation of TrkA EEs and MVBs in distal axons of sympathetic neurons. Interestingly, comparable levels of Rab5$^+$ or CD63-mCherry$^+$ Flag-TrkA endosomes were found in neurons expressing control or Rab7 shRNA, suggesting that Rab7 is not required for formation of TrkA EEs and MVBs in distal axons (*Figure 3—figure supplement 1H–J*).

Next, we sought to determine whether Rab7 is required for TrkA MVBs to become transport competent. We focused on one Rab7 effector protein, RILP, because it mediates late endosome positioning and dynein-dependent movement along microtubules (*Jordens et al., 2001*; *Johansson et al., 2007*). We found that RILP-EGFP was co-localized and co-transported with retrograde Flag-TrkA endosomes in proximal axons of compartmentalized sympathetic neurons (*Figure 3—figure supplement 1K–N*). Moreover, while RILP-EGFP was associated with CD63-mCherry$^+$ Flag-TrkA endosomes in distal axons, Rab7 depletion abolished this association, suggesting that recruitment of RILP to TrkA$^+$ MVBs is Rab7 dependent (*Figure 3—figure supplement 1K*). Moreover, in neurons expressing an shRNA directed against RILP, retrograde TrkA transport was markedly compromised (*Figure 3—figure supplement 1O,P*). Taken together, these findings suggest a Rab7-RILP module associated with TrkA MVBs mediates long-range retrograde transport of TrkA.

## Retrogradely transported TrkA MVBs are signaling competent

In the canonical ligand-receptor endocytic pathway, after receptor tyrosine kinase (RTK) activation and internalization, ligand-receptor complexes follow an endocytic route from early endosomes to MVBs and finally to LEs and lysosomes (*Bergeron et al., 2016*). The prevailing view is that signals from activated RTKs emanate primarily from the inner leaflet of the plasma membrane, and then, following internalization, form early endosomes (*Cosker et al., 2008*; *Harrington and Ginty, 2013*; *Delcroix et al., 2003*; *Sorkin and von Zastrow, 2009*; *Zoncu et al., 2009*). In this model, sorting of RTKs into MVBs is mainly considered in the context of signal deactivation and ligand/receptor degradation. Alternatively, MVBs may function to promote signaling by mediating sorting and degradation of negative regulators (*Taelman et al., 2010*). To ask whether retrogradely transported TrkA$^+$ MVBs have signaling capacity, we monitored their association with effectors of NGF/TrkA signaling. NGF-induced TrkA dimerization and autophosphorylation, particularly on tyrosine residues 490 and 785, signifies receptor activation and functions to trigger downstream signaling cascades, including the Ras/ERK, PLCγ, and PI3K pathways, by recruiting signaling pathway effector proteins to the membrane (*Figure 4—figure supplement 1A*) (*Huang and Reichardt, 2001*; *Sofroniew et al., 2001*; *Watson et al., 2001*). Indeed, both TrkA$^+$ early endosomes and MVBs within axons contain P-TrkA Y490 and Y785, indicative of an active TrkA signaling state (*Figure 4A,B*). Moreover, most TrkA$^+$ MVBs in axons and cell bodies were observed in association with the active form of a key TrkA effector, PLCγ phosphorylated on tyrosine residue 783, suggesting an ability of TrkA MVBs to recruit and activate core TrkA signaling pathway effectors (*Figure 4A,B*; *Figure 4—figure supplement 1B*).

To identify membrane compartments in which retrograde TrkA receptors are capable of signaling, we performed immuno-gold labeling of neurons using the Flag-TrkA retrograde transport assay, in conjunction with antibodies against both P-TrkA and P-PLCγ. Many P-TrkA- and P-PLCγ-labeled structures in cell bodies were observed to be associated with membranes, and the extent of labeling

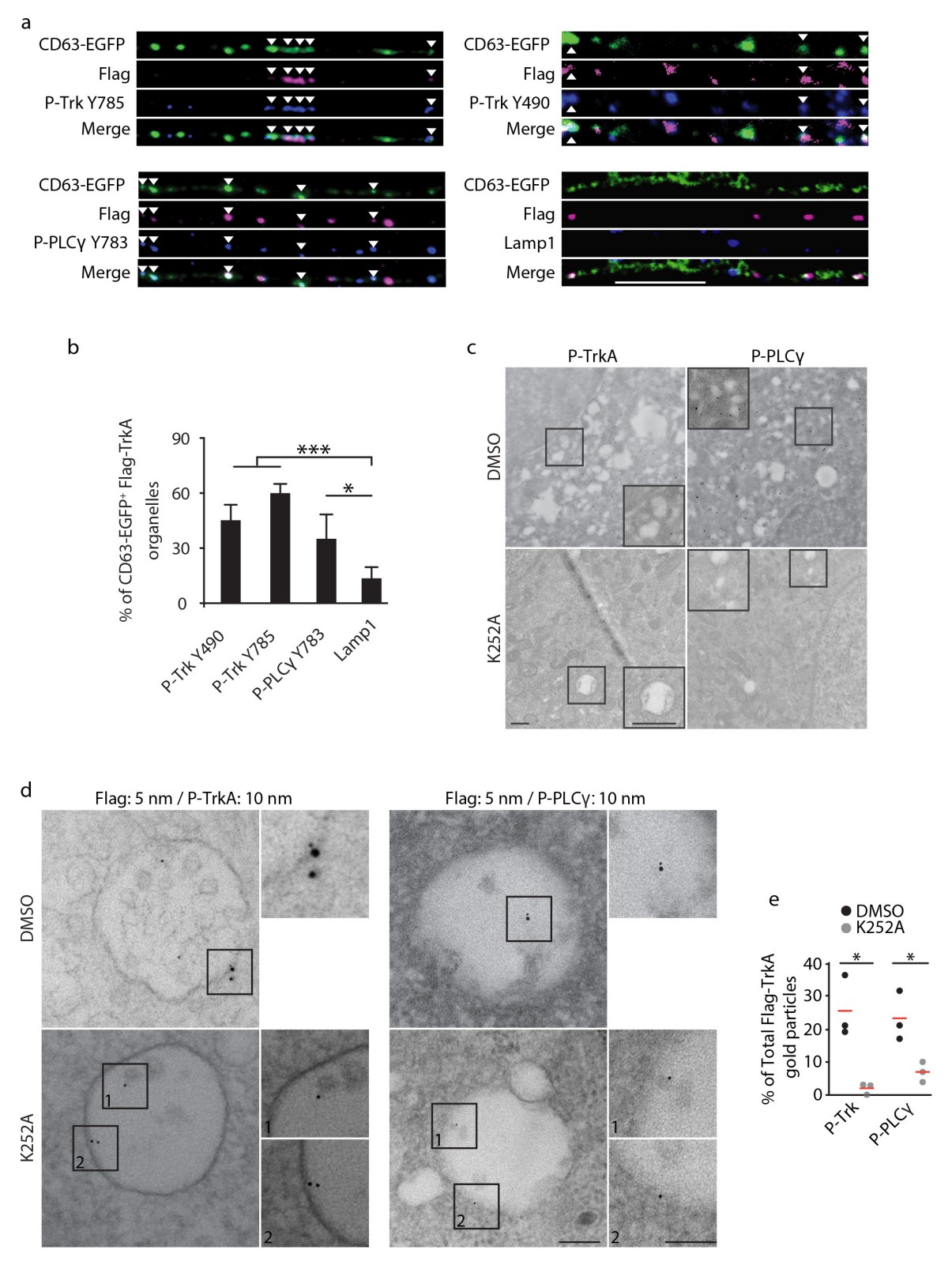

**Figure 4.** Retrograde TrkA[+] MVBs associate with key effectors of the NGF/TrkA signaling pathway. (**a,b**) The Flag-TrkA endosome transport assay was performed in compartmentalized sympathetic neurons expressing CD63-EGFP. The extent of CD63-EGFP[+] Flag-TrkA (green/magenta) punctae in proximal axons colocalized with P-Trk Y490 and Y785, P- PLCγ Y783 and Lamp1 (blue) was quantified (**b**) (n = 3). Scale bar: 10 μm. (**c**) Sympathetic neurons grown in mass culture were NGF-deprived for 16 hr and were then stimulated with NGF for 1 hr in the presence of DMSO (vehicle control) or
*Figure 4 continued on next page*

*Figure 4 continued*

K252a (200 nM), a Trk kinase inhibitor. P-TrkA and P-PLCγ signals in cell bodies were assessed by pre-embed immunogold labeling (n = 3). Insets: high-magnification images of the boxed areas. (d,e) The Flag-TrkA assay was performed in compartmentalized sympathetic neurons using pre-conjugated anti-Flag antibody with a 5-nm gold secondary antibody in the presence of DMSO or K252a in the cell body compartment. Neurons were fixed 1 hr post NGF stimulation and P-TrkA and P-Plcγ signals were revealed by immunogold labeling (10 nm). The extent of Flag-TrkA gold particles associated with P-TrkA or P-PLCγ in cell bodies was assessed (n = 3). Scale bar: 100 nm. Data are represented as mean ± SEM (b) or presented in dot plot (e). *p<0.05 and **p<0.01 by one-way ANOVA with a Tukey's *post-hoc* test (b) or a two tailed unpaired Student's *t* test (e). See also *Figure 4—figure supplement 1*.

DOI: https://doi.org/10.7554/eLife.33012.011

The following figure supplement is available for figure 4:

**Figure supplement 1.** Retrograde TrkA+ endosomes associate with key effectors of NGF/TrkA signaling pathway.

DOI: https://doi.org/10.7554/eLife.33012.012

was greatly diminished following treatment of K252a, a Trk kinase inhibitor, thus confirming the specificity of the immuno-gold labeling strategy (*Figure 4C*). When NGF was applied to cell bodies, co-localization of Flag-TrkA with P-TrkA or P-PLCγ was observed at the plasma membrane as well as associated with early endosomes (*Figure 4—figure supplement 1C* and data not shown). In contrast, when NGF was applied exclusively to distal axons, both P-Trk and P-PLCγ were observed co-localized with retrogradely transported Flag-TrkA on the limiting membrane and intraluminal vesicles (ILVs) of MVBs in cell bodies in a K252a-dependent manner (*Figure 4D,E*), which is consistent with findings of the immunocytochemical experiments. The immediate juxtaposition of the small (5 nm) and large (10 nm) gold particles used to simultaneously visualize Flag-TrkA and P-TrkA, respectively, also confirms that the gold-conjugated anti-Flag antibody remains associated with Flag-TrkA throughout the course of these experiments. We also assessed the extent of co-localization between TrkA+ MVBs and the lysosomal marker Lamp1 in axons and cell bodies. While over 50% of CD63 punctae in both distal and proximal axons were Lamp1+, very few CD63/Flag-TrkA double-labeled punctae were associated with lysosomes in axons and cell bodies (8.5 ± 2.2%; *Figure 4a,b*; *Figure 4—figure supplement 1B*). Taken together, both TrkA+ early endosomes and MVBs recruit NGF/TrkA signaling effector proteins, suggesting that both are capable of TrkA effector signaling, and retrograde TrkA+ MVBs are uniquely not degradative.

## Retrogradely transported TrkA+ endosomes in cell bodies evolve from MVBs into simple, single-membrane vesicle structures and evade lysosomal sorting

From the ultrastructural analysis of retrogradely transported Flag-TrkA+ endosomes in cell bodies, we observed that the majority of TrkA receptors are localized to the membrane of intraluminal vesicles (ILVs) of TrkA+ MVBs while a smaller subset is present on the MVB limiting membrane (*Figure 1*). Based on principles of MVB formation (*Hanson and Cashikar, 2012*; *Katzmann et al., 2002*), previous ultrastructural studies of other RTKs (*Futter et al., 1996*), and our immuno-EM analysis, the topology of TrkA receptors associated with ILVs indicate that its catalytic domain and phosphotyrosine residues are located within the ILV lumen (*Figure 4D*). TrkA receptors in such a configuration are presumably unable to transmit NGF pro-survival signaling cascades to the cytoplasm and nucleus due to the restricted localization of their effector domains. TrkA receptors on the limiting membrane, in contrast, are in a signaling accessible position with their signaling domains facing the cytoplasm. Therefore, we asked whether the membrane localization of TrkA receptors changes to a position favorable for cytoplasmic signaling following MVB entry into cell bodies.

In order to assess the maturation of retrograde TrkA+ MVBs in cell bodies over time, we developed a pulse-block TrkA labeling strategy based on the Flag-TrkA EM assay (*Figure 5A*). For this assay, retrograde TrkA transport was allowed to proceed for 25 min. Then, all subsequent TrkA axonal transport from distal axons into cell bodies was blocked by applying nocodazole to the distal axon compartment to disrupt microtubules, and thus microtubule-dependent axonal transport in that compartment. The distribution of Flag-TrkA receptors in different membrane compartments within cell bodies was then assessed by EM at different time points following nocodazole application. The blockade of retrograde transport in distal axons by nocodazole was highly effective since it completely blocked accumulation of newly formed Flag-TrkA endosomes within the cell body

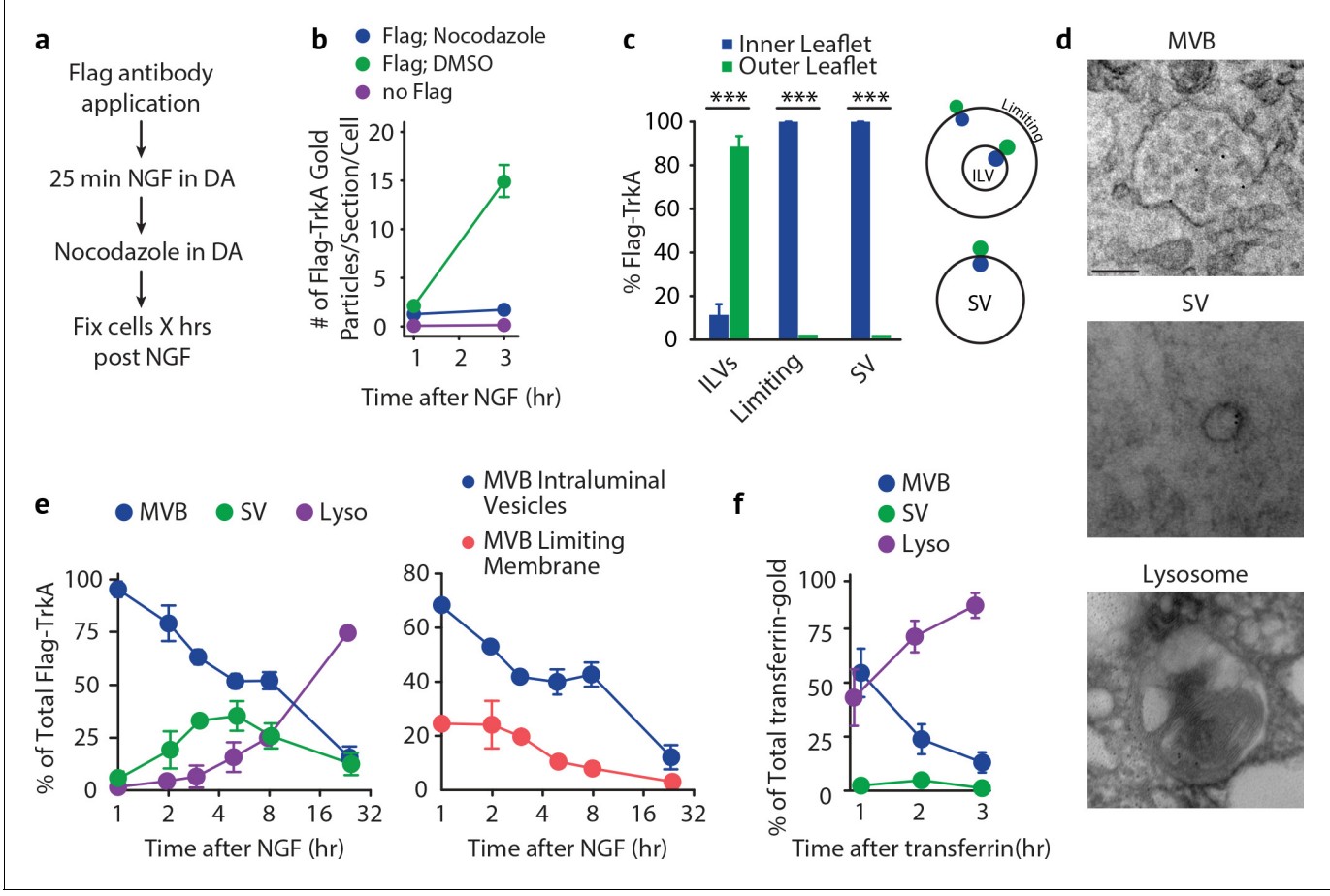

**Figure 5.** Retrogradely transported TrkA⁺ endosomes within cell bodies evolve from MVBs into simple, single-membrane vesicle structures. (a) Schematic of the pulse-block assay. The Flag-TrkA assay is performed in *Ntrk1^Flag* sympathetic neurons cultured in compartmentalized microfluidic chambers as in *Figure 1—figure supplement 1A* using pre-conjugated anti-Flag antibody and Protein A-5 nm gold. Nocodazole is applied to distal axons 25 min post-NGF application to block retrograde transport. Neurons are fixed at indicated time points and processed for EM. (b) The Flag-TrkA assay was performed in WT neurons, *Ntrk1^Flag* neurons treated with either DMSO or nocodazole in distal axons (25 min post NGF). Cells were fixed at indicated time points and processed for EM. The number of gold particles per section per cell was quantified (n = 3, Results were pooled from over 150 Flag-gold particles). (c) The pulse-block assay was performed as in (a), and membrane topology of the Flag epitope was assessed. A schematic of the inner- or outer-leaflet position for membrane of ILVs, the limiting membrane and SVs is shown on the right. (d,e) The pulse-block assay was performed and the distribution of retrogradely transported Flag-TrkA gold particles in MVBs, single-membrane vesicle structures (SVs) and lysosomes within the cell body over time was assessed (e) Results were pooled from over 150 Flag-gold particles from four samples for each time point. Representative images of retrograde Flag-TrkA in each of the three membrane compartments are shown in (d). (f) Sympathetic neurons in compartmentalized cultures were incubated with transferrin-gold (6 nm) in distal axons and the pulse-block assay was performed. The distribution of retrograde transferrin-gold in MVBs, SVs and lysosomes was assessed by EM. For (d–f), over 150 endosomes were counted for each condition at each time point in four independent experiments. Scale bar: 100 nm. Data are represented as mean ± SEM. ***p<0.001 by two-way ANOVA with a Tukey's *post-hoc* test (e) or a two-tailed unpaired Student's *t* test (c). See also *Figure 5—figure supplement 1*.

DOI: https://doi.org/10.7554/eLife.33012.013

The following figure supplement is available for figure 5:

**Figure supplement 1.** Nocodazole treatment effectively blocks microtubule-dependent axonal trafficking and retrograde Flag-TrkA transport.

DOI: https://doi.org/10.7554/eLife.33012.014

compartment (*Figure 5—figure supplement 1B*). In addition, live cell imaging experiments revealed that retrograde axonal movement of Flag-TrkA⁺ endosomes in distal axons ceased within 15 to 20 min of nocodazole application (*Figure 5—figure supplement 1C*). Moreover, nocodazole treatment at 25 min post-NGF application to distal axons resulted in sparse accumulation of retrograde TrkA endosomes in cell bodies at the 1 hr time point post-NGF treatment and a relatively low, but stable number of TrkA receptors were observed over the subsequent several hours (*Figure 5B*). In contrast,

neurons without axonal transport blockade exhibited a dramatic increase in the number of retrograde Flag-TrkA$^+$ vesicles appearing in cell bodies during the 1 hr time period (*Figure 5B*). Furthermore, nocodazole application to distal axons did not affect microtubule-dependent endosome maturation and trafficking within the cell body compartment (*Figure 5—figure supplement 1D,E*), indicating effective compartmentalization of transport blockade. These findings indicate that any change in TrkA$^+$ endosome ultrastructure in cell bodies at times following addition of nocodazole to distal axons is the result of maturation or evolution of existing endosomes that had arrived prior to nocodazole application, and not newly trafficked endosomes. Thus, this pulse-block Flag-TrkA EM paradigm allowed us to define the dynamics of membrane localization of retrogradely transported TrkA receptors following their arrival to the cell body.

Using this pulse-block TrkA labeling assay, the majority of TrkA receptors observed in cell bodies by EM were found to be associated with one of three types of membrane structures: MVBs, single-membrane vesicles (SVs), and lysosomes (*Figure 5D*). Moreover, the membrane localization of the Flag epitope was in agreement with the principle of endocytosis: the Flag epitope was found on the outer leaflet of ILV membranes and the inner leaflet of limiting membranes and SVs (*Figure 5C* and *Figure 1B*). Consistent with our previous experiments, 1 hr after NGF application to distal axons, and following blockade of newly arriving endosomes, the majority (92.8 ± 3.6%) of retrograde TrkA$^+$ endosomes were MVBs (*Figure 5E*, left panel). Among the MVB-resident TrkA receptors, 68.3 ± 2.6% were localized to ILVs and 24.5 ± 4.7% to the limiting membrane (*Figure 5E*, right panel). Strikingly, the fraction of total cell body Flag punctae found associated with MVBs decreased during the next several hours, and this decrease was coincident with the emergence of Flag-TrkA$^+$ SVs, which peaked (42.4 ± 8.2%) at ~5 hr post-NGF treatment (*Figure 5E*). Moreover, the fraction of Flag-TrkA$^+$ puncta associated with lysosomes remained low during the first 8 hr of blocking the arrival of new endosomes, and by 24 hr post-NGF the majority of Flag-TrkA receptors were found associated with lysosomes (*Figure 5E*).

Because the transport blockade prevented TrkA$^+$ endosomes from entering cell bodies in this pulse-block paradigm, the most parsimonious explanation of the source of the TrkA$^+$ SV population during the course of these experiments is that they emerge de novo within cell bodies. Since the vast majority of newly arrived Flag-TrkA$^+$ endosomes at the start of the transport blockade are MVBs, the Flag-TrkA$^+$ SVs are therefore derived, either directly or indirectly, from TrkA$^+$ MVBs. These observations indicate that the membrane localization of retrogradely transported TrkA, and indeed the ultrastructural properties of TrkA$^+$ endosomes, changes following TrkA$^+$ endosome entry into the cell body.

To ask whether the MVB-to-SV transition and delayed lysosomal sorting occurs for MVBs carrying other types of cargoes, we next assessed the fate of retrogradely transported endosomes carrying transferrin, a 'non-signaling' cargo. Transferrin was internalized in distal axons in a dynamin-dependent manner and colocalized with the transferrin receptor (*Figure 6—figure supplement 1A,B*). The majority of newly internalized transferrin and Flag-TrkA were co-localized within distal axons, suggesting that they can be sorted into the same early endosomes (*Figure 6—figure supplement 1C*). Interestingly, the level of co-localization decreased over time, indicating that these cargoes were later sorted into different vesicular compartments (*Figure 6—figure supplement 1C*). Moreover, retrogradely transported Flag-TrkA and transferrin remained segregated upon their arrival into cell bodies (*Figure 6—figure supplement 1D*). Consistent with this, in EM experiments in which transferrin and Flag-TrkA were labeled with different size gold particles, we found that retrogradely transported transferrin and TrkA were localized to distinct MVBs (*Figure 6—figure supplement 1E*). In addition, in contrast to the observed emergence of TrkA$^+$ SVs in cell bodies, we did not observe SVs containing transferrin (*Figure 5F*). Instead, the majority of retrogradely transported transferrin proteins were observed in lysosomes, beginning as early as the 2 hr time point, when TrkA was mostly associated with MVBs and SVs. Similar results were found with MVBs containing Cholera toxin subunit B (CTB) or Bovine serum albumin (BSA) (See below, Figure 7B,C; *Figure 7—figure supplement 1F,G*). Therefore, transferrin and TrkA are retrogradely transported in distinct MVBs populations, and TrkA$^+$ MVBs are unique in that they give rise to SVs and evade lysosomal fusion.

## TrkA⁺ single vesicles in cell bodies are signaling competent and not associated with the early endosome marker Rab5

We next asked whether TrkA$^+$ SVs that emerge in cell bodies are poised for signaling and associated with signaling effectors. SVs were visualized using immuno-gold labeling of P-TrkA and P-PLCγ in the pulse-block Flag-TrkA EM paradigm. Indeed, 5 hr following NGF application to distal axons, Flag-TrkA receptors localized to the membrane of SVs were observed co-localized with P-TrkA and P-PLCγ, with the Flag epitope oriented on the luminal side of the SV membrane and the phospho-tyrosine residues on the cytoplasmic side (*Figure 6A*). These observations suggest that TrkA$^+$ SVs derived from MVBs are signaling competent and capable of transducing TrkA signals within the cell body.

We also sought to define molecular features of TrkA$^+$ SVs in cell bodies. To determine whether TrkA$^+$ SVs in cell bodies of sympathetic neurons are associated with Rab5 following retrograde transport, we assessed the extent of co-localization between Flag-TrkA and Rab5 in cell bodies (*Figure 6B*). At the 5 hr time point of the pulse-block Flag-TrkA assay, when approximately 40% of retrogradely transported Flag-TrkA$^+$ endosomes are of a single vesicular nature (*Figure 5E*), fewer than 5% of the Flag punctae in cell bodies were co-localized with Rab5 (*Figure 6B,D*). The extent of co-localization of Flag-TrkA and Rab5 was also assessed by EM (*Figure 6C*). To visualize Rab5$^+$ endosomes by EM, we fused Rab5 with APEX2 (*Figure 6—figure supplement 1F*), a modified per-oxidase that enables visualization of the product of the peroxidase-catalyzed reaction by EM and thus subcellular distribution of APEX2 fusion proteins (*Lam et al., 2015*). Lentivirus-mediated deliv-ery of APEX2-Rab5 into sympathetic neurons and EM analysis revealed that the majority of newly internalized Flag-TrkA receptors are associated with APEX2-Rab5$^+$ endosomes (67.9 ± 7.2%; *Figure 6C*, middle; *Figure 6E*), as predicted. In contrast, and consistent with our immunocytochemi-cal analyses, very few Flag-TrkA punctae were found to be associated with APEX2-Rab5$^+$ structures at the 5 hr time point of the pulse-block Flag-TrkA assay (6.4 ± 2.0%; *Figure 6C*, right; *Figure 6E*). These findings indicate that TrkA$^+$ SVs that emerge from MVBs in cell bodies are not associated with Rab5.

Endocytic pathways downstream of MVBs, other than the well-described degradation pathway, include the recycling pathway and the retromer-dependent transport pathway to the Golgi appara-tus (*Katzmann et al., 2002*; *Cullen and Korswagen, 2011*). Since TrkA$^+$ SVs in cell bodies are derived from TrkA$^+$ MVBs, the TrkA endocytic pathway in cell bodies might engage one or both of these routes. Indeed, we observed some TrkA $^+$ SVs in close proximity to Golgi, while others were close to the plasma membrane and exhibited a tubular shape resembling recycling endosomes (*Fig-ure 6—figure supplement 1H*). To more directly test these possibilities, we assessed the extent of Flag-TrkA co-localization with Rab11 and Vps35, key components of recycling endosomes and the retromer complex, respectively (*Figure 6F*). At the 5 hr time point of the pulse-block Flag-TrkA assay, Flag-TrkA$^+$ endosomes were associated with Vps35 (39.8 ± 12.5%) and to a lesser extent Rab11 (13.0 ± 6.6%, *Figure 6D,F*). Interestingly, nearly all Rab11$^+$ Flag-TrkA endosomes were also Vps35$^+$(10.3 ± 5.7%, *Figure 6D,F*). We confirmed the Vps35 identity of TrkA$^+$ SVs by APEX2-Vps35/Flag EM double labeling (58.9 ± 8.3% of total Flag-TrkA SVs, *Figure 6G*; *Figure 6—figure supple-ment 1G*). Therefore, a substantial portion of TrkA$^+$ SVs exhibit a molecular feature of retromer com-plexes, and a subset may include Rab11$^+$ recycling endosomes.

## TrkA kinase activity directs the fate of retrogradely transported TrkA⁺ endosomes

How are MVB, SV and lysosome membrane dynamics regulated following arrival of TrkA$^+$ MVBs in cell bodies, and to what extent does TrkA signaling control this process? To address this, we used a chemical genetic strategy to manipulate TrkA kinase signaling in a compartmentalized manner. For this purpose, we generated a Flag-TrkB/A-F592A expression construct (*Figure 7—figure supple-ment 1A*). This construct encodes a chimeric receptor consisting of the TrkB extracellular domain and the TrkA transmembrane and intracellular domains so that BDNF rather than NGF can activate TrkA intracellular signaling in transduced cells (*Figure 7—figure supplement 1A*). Moreover, the TrkB/A fusion protein contains the Flag epitope tag for detecting retrogradely transported receptors by both light microscopy and EM, as done using neurons harvested from *Ntrk1$^{Flag}$* mice. In addition, the construct contains the F592A point mutation within the TrkA kinase domain ATP binding pocket,

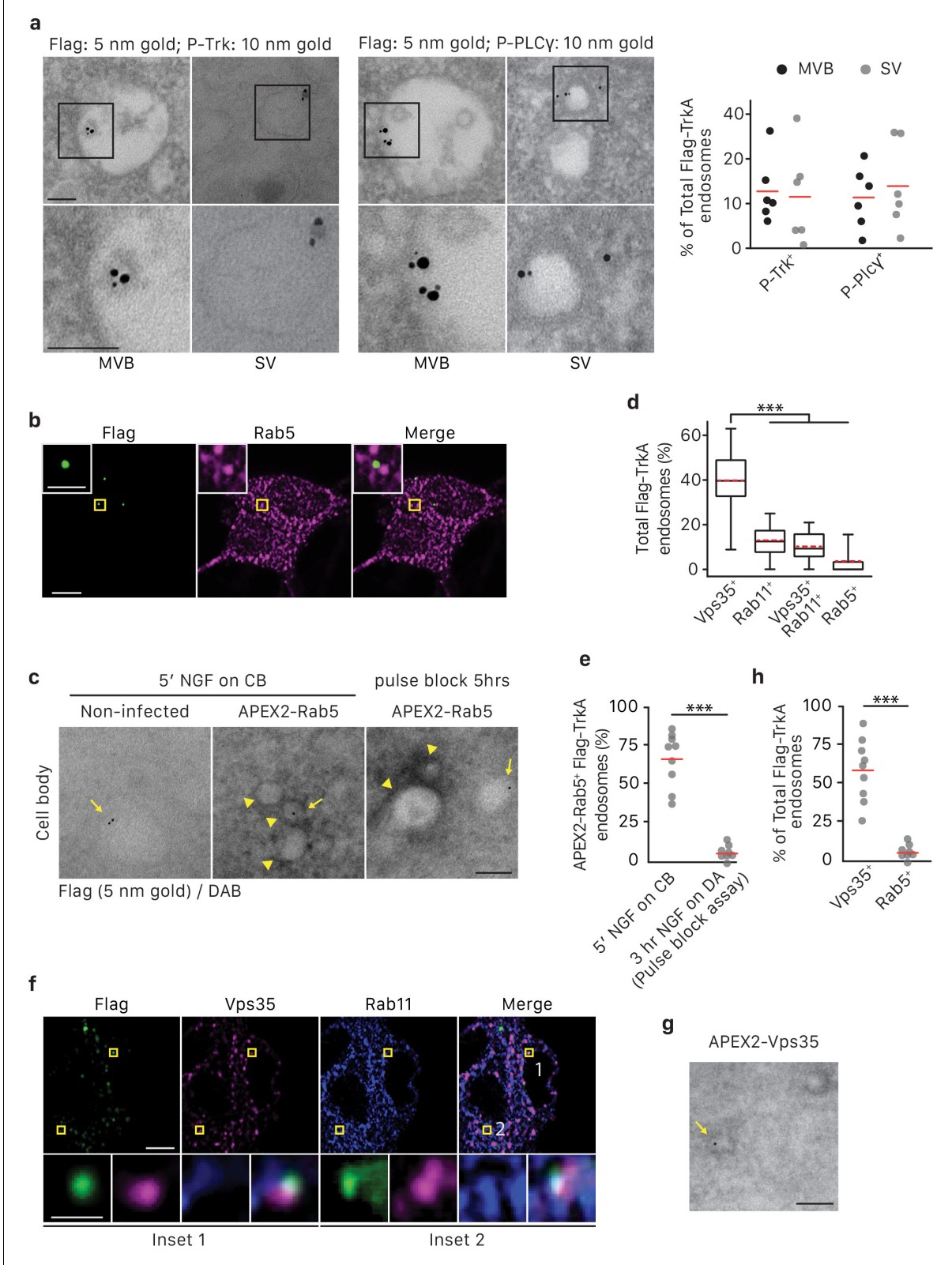

**Figure 6.** TrkA[+] single vesicles formed de novo in cell bodies after retrograde transport are signaling competent and are not Rab5[+]early endosomes. (a) The pulse-block assay was performed in compartmentalized sympathetic neurons using pre-conjugated anti-Flag antibody with 5 nm gold secondary antibody. Cells were fixed 5 hr post-NGF application and immunogold-labeled for P-Trk or P-PLCγ (10 nm). Shown are representative images of an MVB and SV containing retrograde Flag-TrkA gold particles that are juxtaposed to P-Trk or P-PLCγ. High magnification images of the boxed areas are

*Figure 6 continued on next page*

*Figure 6 continued*

shown in the bottom panel. The percentage of these signaling competent MVBs and SVs in cell bodies was quantified (n = 6). Scale bar: 100 nm. (**b,d**) The pulse-block assay was performed in compartmentalized sympathetic neurons and colocalization between Flag-TrkA (green) and Rab5 (magenta) in cell bodies was assessed 5 hr post-NGF application. Insets show magnification of the boxed areas. Quantification is shown in (**d**) (n = 4). Scale bar: 5 μm and 1 μm (inset). (**c,e**) Sympathetic neurons grown in mass culture were infected with a virus expressing APEX2-Rab5. The Flag-TrkA assay was performed using pre-conjugated anti-Flag antibody with Protein A-5 nm gold secondary antibody and cells were fixed 5 min post-NGF stimulation. DAB staining was performed and cells were processed for EM. The extent of gold particles associated with APEX2$^+$ endosomes in cell bodies was assessed. APEX2$^+$ endosomes were identified based on the dark staining associated with endosomal membranes (arrowheads), compared to the lack of contrast in non-infected cells (left panel). In the right panel, the pulse-block assay was performed in compartmentalized sympathetic neurons expressing APEX2-Rab5. Cells were fixed 5 hr post-NGF stimulation and the extent of gold particles resided in APEX2$^+$ endosomes was assessed. Arrows denote endosomes containing Flag-TrkA and arrowheads denote APEX2$^+$ endosomes. Quantification is shown in (**e**) (n = 3). Scale bar: 100 nm. (**f**) The pulse-block assay was performed in compartmentalized sympathetic neurons and colocalization between Flag-TrkA (green), Vps35 (magenta) and Rab11 (blue) in cell bodies was assessed at 5 hr post-NGF application. Insets show magnification of the boxed areas. Quantification is shown in (**d**) (n = 4). Scale bar: 5 μm and 1 μm (inset). (**g**) The pulse-block assay was performed in compartmentalized sympathetic neurons expressing APEX2-Vps35. Cells were fixed 5 hr post-NGF stimulation and the extent of gold particles resided in APEX2$^+$ endosomes was assessed (n = 3). Shown is a representative EM image. Scale bar: 100 nm. Data are presented in dot plot (**a,e,h**) or box plot (**d**). \*\*\*p<0.001 by one-way ANOVA with a Tukey's *post-hoc* test (**d**) or a two-tailed unpaired Student's *t* test (**e,g**). See also *Figure 6—figure supplement 1*.
DOI: https://doi.org/10.7554/eLife.33012.015

The following figure supplement is available for figure 6:

**Figure supplement 1.** Retrogradely transported transferrin and Flag-TrkA are sorted in distinct MVBs and characterization of TrkA$^+$ single vesicles.
DOI: https://doi.org/10.7554/eLife.33012.016

which renders TrkA catalytic activity sensitive to inhibition by the small molecule 1NMPP1 (*Figure 7—figure supplement 1A*) (*Chen et al., 2005*).

As expected (*Atwal et al., 2000*), expression of Flag-TrkB/A-F592A enabled survival of sympathetic neurons treated with BDNF acting exclusively on distal axons (*Figure 7—figure supplement 1C*). Furthermore, exposure of Flag-TrkB/A-F592A infected neurons to BDNF in the absence of NGF was sufficient to promote the generation and transport of Flag-TrkB/A-F592A$^+$ signaling endosomes, as seen by an increase in P-Trk immunoreactivity by both light microscopy and EM (*Figure 7—figure supplement 1D,E*). Importantly, Flag-TrkB/A-F592A autophosphorylation was eliminated upon exposing neurons to 1NMPP1 (*Figure 7—figure supplement 1D,E*). The elimination of P-Trk signaling in the context of BDNF stimulation also confirmed the lack of activation of endogenous TrkA receptors, which are insensitive to 1NMPP1. Therefore, the Flag-TrkB/A-F592A construct enables both immunodetection of retrogradely transported Flag-TrkB/A-F592A$^+$ endosomes and precise temporally and spatially restricted manipulation of its associated TrkA kinase activity.

To test whether TrkA kinase activity is required for membrane redistribution of retrogradely transported TrkA receptors following TrkA$^+$ endosome entry into cell bodies, we used the pulse-block Flag-TrkA assay and sympathetic neurons that express either Flag-TrkB/A-F592A or Flag-TrkB/A-WT (lacking 1NMPP1 sensitivity) and treated with either 1NMPP1 or the vehicle DMSO applied exclusively to the cell body compartment (*Figure 7—figure supplement 1B*). Distal axons of these neurons were exposed to BDNF for 5 hr, and the extent of lysosomal localization of retrogradely transported receptors was then assessed by Flag/Lamp1 co-localization (*Figure 7A*). Interestingly, in cells expressing Flag-TrkB/A-F592A and treated with 1NMPP1 applied to their cell bodies, we observed a dramatic increase in Lamp1$^+$ Flag endosomes compared to the three control conditions (*Figure 7A,B*). These findings indicate that TrkA kinase signaling in cell bodies prevents precocious sorting of retrograde TrkA$^+$ endosomes to lysosomes.

To further explore the role of TrkA kinase activity in endosome sorting within cell bodies, we analyzed the relative amounts of retrogradely transported Flag-TrkB/A-F592A in association with MVBs, SVs and lysosomes by EM (*Figure 7B,C*). Neurons expressing Flag-TrkB/A-F592A and treated with either DMSO or 1NMPP1 in the cell body compartment exhibited a similar pattern of TrkA membrane localization at the 1 hr time point, with the majority of Flag-TrkB/A-F592A associated with MVBs (*Figure 7C*). However, at later time points, neurons exposed to 1NMPP1 on cell bodies exhibited a much lower fraction of Flag-TrkB/A-F592A$^+$ MVBs, virtually no Flag-TrkB/A-F592A$^+$ SVs, and a much higher percentage of Flag-TrkB/A-F592A$^+$ lysosomes, compared to DMSO control-treated neurons (*Figure 7B,C*). These observations indicate that TrkA kinase activity is necessary for the

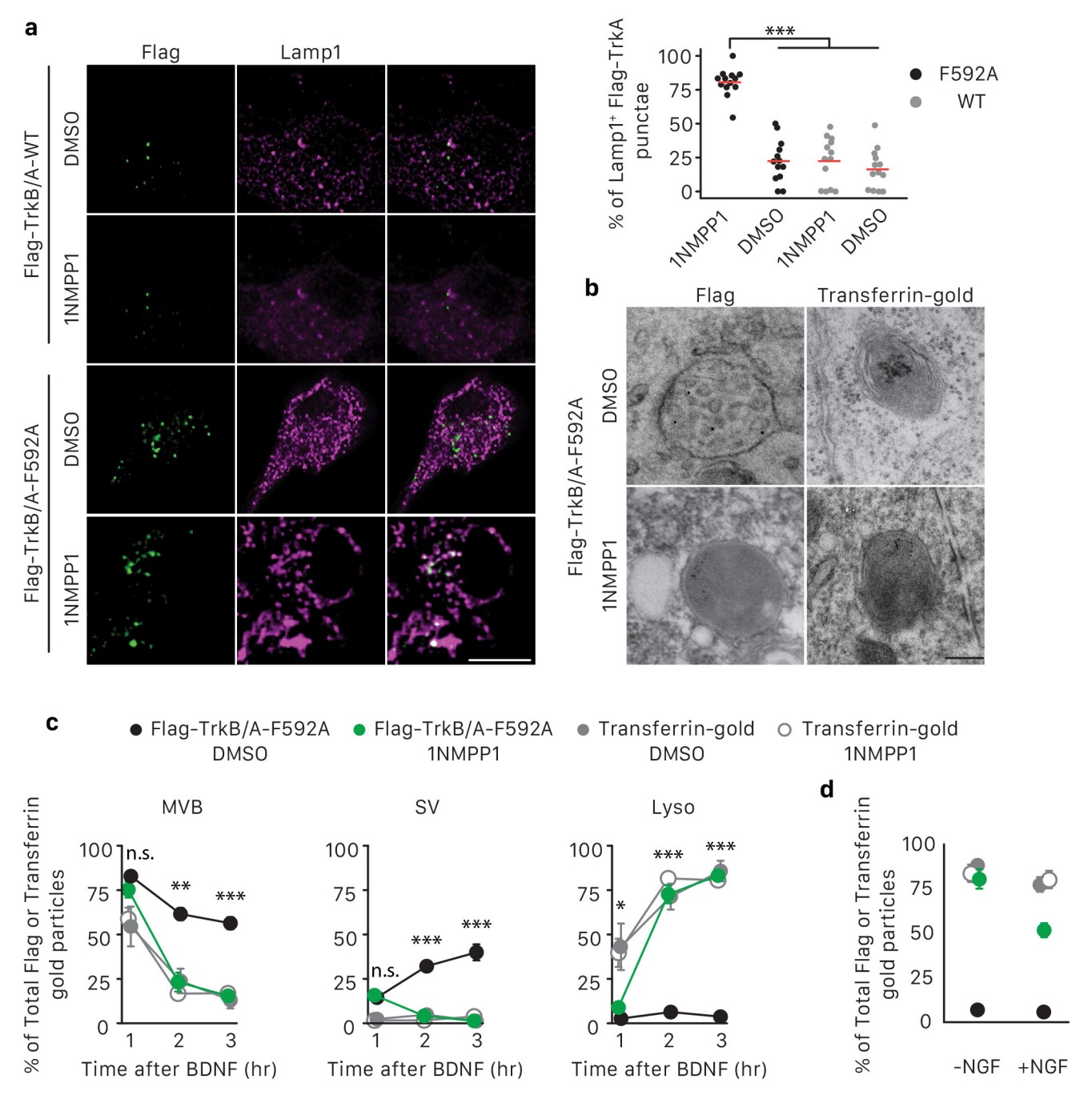

**Figure 7.** TrkA kinase activity within endosomes regulates maturation and fate of retrograde TrkA[+] endosomes. (a) Compartmentalized WT sympathetic neurons were infected with a virus expressing either Flag-TrkB/A-WT or Flag-TrkB/A-F592A. The pulse-block kinase assay was performed as in *Figure 5a*, with BDNF instead of NGF stimulation in distal axons and with the cell body compartment treated with DMSO or 500 nM 1NMPP1 during the course of the experiments. Cells were fixed 5 hr post-BDNF application, and colocalization between Flag-TrkA (green) and the lysosome marker Lamp1 (magenta) was assessed (n = 3). Scale bar: 10 μm. (b,c) The pulse-block kinase assay was performed in sympathetic neurons expressing Flag-TrkB/A-F592A using pre-conjugated anti-Flag antibody with Protein A-5 nm gold or a transferrin-gold tracer (6 nm). Cells were fixed at indicated time points and the number of Flag-TrkB/A-F592A or transferrin-gold particles associated with MVBs, SVs and lysosomes was scored (c) (n = 3). Shown in (b) are representative EM micrographs of retrogradely transported Flag-TrkA and transferrin-gold in MVBs or lysosomes at the 3 hr time point. Scale bar: 100 nm. (d) The pulse-block kinase assay was performed as in (b) with either the presence or absence of NGF in the cell body compartment during the course of experiments. The extent of lysosomal Flag-TrkA or transferrin-gold was assessed 3 hr post-BDNF application (n = 3). Data are presented

*Figure 7 continued*

in dot plot (a) or represented as mean ± SEM (c,d). *p<0.05, **p<0.01 and ***p<0.001 by two-way ANOVA with Tukey's *post-hoc* test (a) or three-way ANOVA with Tukey's *post-hoc* test (c,d). See also *Figure 7—figure supplement 1*.

DOI: https://doi.org/10.7554/eLife.33012.017

The following figure supplement is available for figure 7:

**Figure supplement 1.** Expression of Flag-TrkB/A-F592A allows specific activation and inhibition of TrkA kinase activity within endosomes.

DOI: https://doi.org/10.7554/eLife.33012.018

proper distribution of retrogradely transported TrkA within MVBs and SVs and the prevention of premature lysosomal sorting following endosome arrival to cell bodies.

Finally, we asked whether the distribution of TrkA in MVBs, SVs and lysosomes is controlled by local TrkA signaling associated with endosomes themselves or whether TrkA signals emanating from other platforms, that is in trans from other TrkA endosomes or the plasma membrane, can control the fate of retrogradely transported TrkA endosomes. For this, we compared the fate of retrogradely transported endosomes that contain either TrkA or non-TrkA cargoes, including transferrin, bovine serum albumin (BSA), or Cholera toxin subunit B (CTB), by EM analysis (*Figure 7B,C*; *Figure 7—figure supplement 1F,G*). As for TrkA$^+$ endosomes, the majority of retrogradely transported transferrin$^+$, BSA$^+$ or CTB$^+$ endosomes newly arrived within cell bodies were MVBs (*Figure 7C*; *Figure 7—figure supplement 1F,G*). In dramatic contrast to TrkA$^+$ endosomes, however, very few, if any transferrin$^+$, BSA$^+$ and CTB$^+$ endosomes redistributed to SVs during the subsequent several hours. Rather, the vast majority of these non-TrkA MVBs were rapidly sorted into lysosomes (*Figure 7B,C*; *Figure 7—figure supplement 1F,G*). This rapid sorting of transferrin$^+$, BSA$^+$ and CTB$^+$ endosomes to lysosomes was independent of 1NMPP1 treatment, and was similar to the rapid sorting of TrkA$^+$ endosomes to lysosomes observed in neurons expressing Flag-TrkB/A-F592A and receiving 1NMPP1 treatment. Lastly, we asked whether TrkA signaling acting in trans can control the fate of retrogradely transported TrkA$^+$ MVBs. To address this possibility, we conducted similar Flag-TrkB/A-F592A experiments as those described above, although these were performed in the presence of NGF applied exclusively to the cell body compartment to activate endogenous TrkA signaling. Interestingly, NGF application to the cell body compartment partially rescued premature lysosome sorting of retrograde Flag-TrkB/A-F592A$^+$ MVBs in neurons treated with 1NMPP1 applied to the cell bodies (*Figure 7D*). In contrast, NGF application directly to cell bodies did not prevent the rapid sorting of retrogradely transported transferrin-containing endosomes to lysosomes (*Figure 7D*). Thus, the fate of retrogradely transported MVBs within cell bodies is dependent on the nature and signaling activity of its cargo. TrkA kinase signaling promotes the emergence of TrkA$^+$ SVs from TrkA$^+$ MVBs and protects or delays these endosomes from acquiring a lysosomal fate. Moreover, TrkA receptors can signal in trans, albeit to a lesser extent than endosomal TrkA signals acting in cis, to influence the fate of retrograde TrkA$^+$ signaling endosomes and prevent their lysosomal targeting and degradation.

## Discussion

The type of endosome that mediates long-range retrograde neurotrophin signaling has been controversial. Here, we report that in both sympathetic neurons and TrkA$^+$ primary sensory neurons, the majority of retrograde TrkA$^+$ endosomes are associated with MVB/late endosome proteins, including Rab7, CD63 and Hrs, and that Rab7 mediates retrograde TrkA transport and signaling both in vitro and in vivo. Our ultrastructural analysis revealed that retrogradely transported TrkA receptors are carried predominantly in MVBs in a signaling competent state. Moreover, upon their arrival into cell bodies, TrkA$^+$ MVBs give rise to a previously undescribed type of signaling competent TrkA$^+$ single vesicle, a subset of which are associated with the retromer complex component Vps35. Thus, MVBs mediate long-range retrograde axonal TrkA transport and retrograde TrkA signaling in neuronal cell bodies (*Figure 8*).

Our findings are most consistent with earlier in vitro and in vivo studies in which radiolabeled NGF was found in MVB/lysosomal structures in axons and cell bodies of sympathetic neurons (*Sandow et al., 2000*; *Bhattacharyya et al., 2002*). Although those observations implicated MVBs in mediating axonal transport of NGF, it had been unclear whether MVBs carry active TrkA receptors

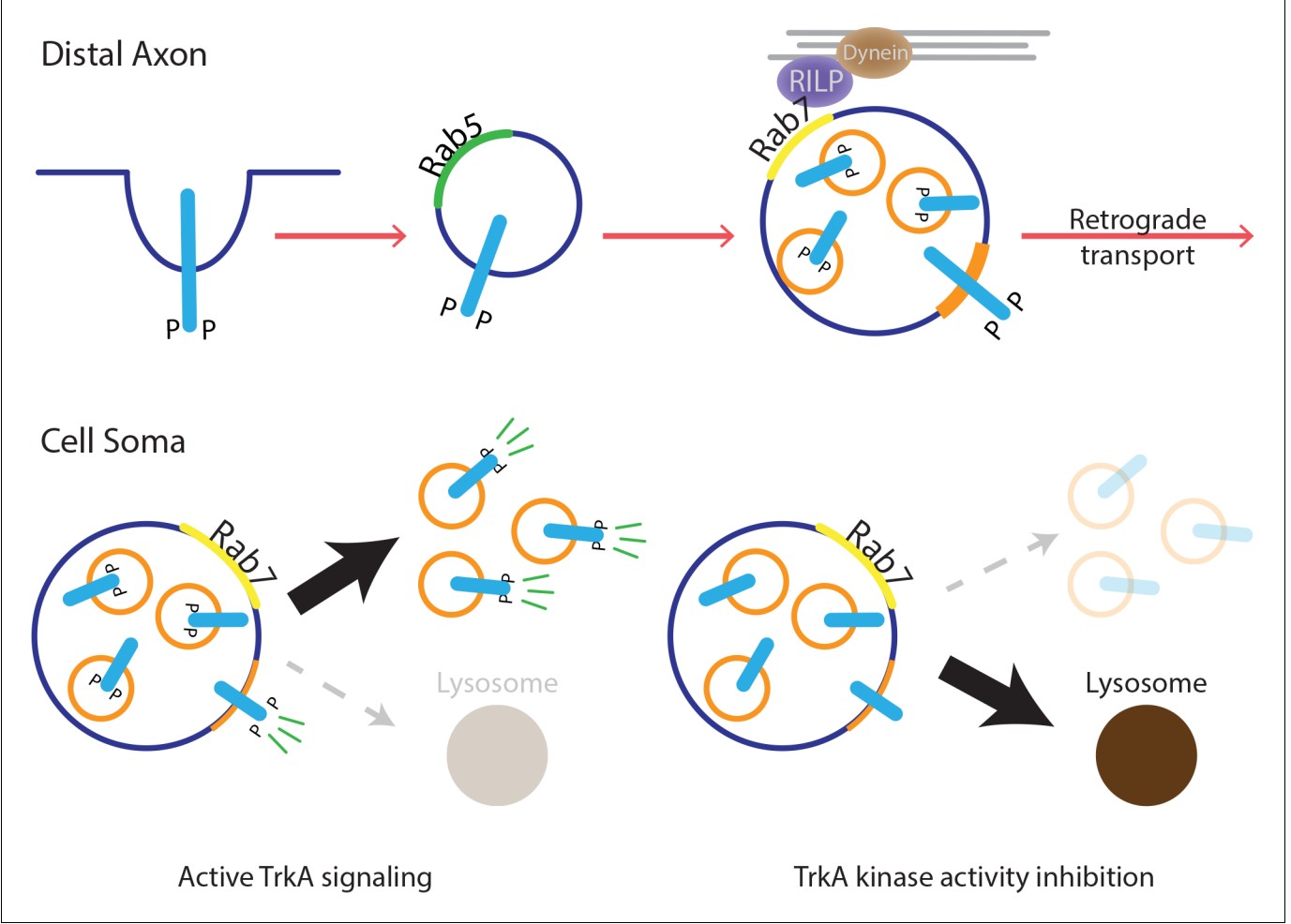

**Figure 8.** Model for MVB-mediated retrograde TrkA transport and signaling. This schematic illustrates a model for how multivesicular bodies control propagation of retrograde NGF-TrkA signals. In distal axons, newly internalized NGF/TrkA complexes are sorted into Rab5$^+$ early endosomes, and these TrkA$^+$ early endosomes mature to form multivesicular bodies. Rab7, which localizes to TrkA$^+$ MVBs, facilitates rapid long-range retrograde axonal transport, and in turn, neuronal survival and synapse formation. In neuronal soma, active endosomal TrkA signaling induces de novo formation of single-membrane vesicles from retrogradely transported MVBs and prevents TrkA$^+$ MVBs fusion with lysosomes. Together, these TrkA activity-dependent MVB dynamics promote and sustain transduction of retrograde NGF signals.

DOI: https://doi.org/10.7554/eLife.33012.019

derived from distal axons since NGF can also bind to p75$^{NTR}$, and NGF may be separated from TrkA and p75$^{NTR}$ following internalization. As such, whether MVBs transmit essential retrograde NGF signals within cell bodies was unknown. Our Flag-TrkA labeling strategy provided a means to visualize retrograde TrkA endosomes, and Flag/P-Trk and Flag/P-PLCγ double immuno-EM experiments revealed a signaling competent state of retrograde TrkA$^+$ MVBs in cell bodies. This observation is in agreement with a previous study in which P-Trk immunoreactivity was found associated with multivesicular structures in the sciatic nerve (*Bhattacharyya et al., 2002*). Other studies have suggested a lack of association of BDNF with MVBs or other endosomal structures in the hypoglossal motor neurons in vivo (*Altick et al., 2009*) and embryonic stem-cell-derived motor neurons in vitro (*Terenzio et al., 2014*), and this discrepancy may be the result of low efficiency of labeling or that at least some other cell types utilize different mechanisms for retrograde BDNF transport. Future work will be needed to address whether BDNF or other neuronal growth factors and neuron types use mechanisms similar to those reported here for NGF in sympathetic and sensory neurons.

The present findings cause us to reconsider a long-held view in neuronal cell biology, that Rab5$^+$ early endosomes mediate long-range propagation of target-derived neurotrophic factor signals from

distal axons to cell bodies. Prior evidence supporting the early endosome model included the finding of co-localization of NGF and Rab5 in neuronal soma in vivo and the presence of TrkA in Rab5$^+$ endosomes within cytoplasmic contents eluted from proximal segments of an ex vivo sciatic nerve preparation following NGF injection into the periphery (*Delcroix et al., 2003*). However, the lack of means for specific labeling of distal axon-derived TrkA$^+$ endosomes may complicate interpretations of those findings because of the possibility that the observed Rab5$^+$ structures were formed locally at the plasma membrane of the axon or cell body and therefore not representative of retrogradely transported TrkA$^+$ endosomes. On the other hand, we cannot exclude the possibility that early endosomes carry a small subset of retrograde NGF signals, because ~7% of retrograde TrkA$^+$ endosomes observed in the pulse block experiments at the earliest time point following transport blockade are single-membrane vesicles (*Figure 5E*). Nevertheless, considering the differences between the quantity of TrkA$^+$ MVBs and TrkA$^+$ early endosomes, the amount of TrkA receptors carried per endosome, and differences in MVB and early endosome movement properties, MVBs may provide a favorable means of long-range TrkA translocation and signal propagation, compared to early endosomes. It is noteworthy that, like TrkA, transferrin, BSA and CTB internalized in distal axons were also retrogradely transported to cell bodies mainly, if not entirely, via MVBs, suggesting to us that MVBs are the predominant and possibly sole retrograde axonal transport carrier of cargoes internalized in distal axons (*Figure 7C*; *Figure 7—figure supplement 1G*).

Our in vivo findings demonstrate that Rab7 participates in TrkA signaling, survival and synaptogenesis of sympathetic neurons. In vitro, Rab7 is associated with, and necessary for retrograde transport of TrkA$^+$ signaling endosomes. Moreover, Rab7 is required for neuronal survival when NGF is applied exclusively to distal axons. The sympathetic neuron survival defect in Rab7 mutant SCGs in vivo is, however, considerably less dramatic than that in SCGs of either *Ntrk1* or *Ngf* null mice in vivo (*Smeyne et al., 1994*; *Crowley et al., 1994*). This discrepancy may be attributed to the incomplete ablation of *Rab7* between E14 and P0 (*Figure 3—figure supplement 1A*). Alternatively, a fraction of NGF survival signaling may occur via Rab7-independent MVB transport mechanism, NGF acting directly on cell bodies, or through early endosomes or another retrograde signal carrier that compensates for loss of Rab7 at embryonic times. Although a more complete depletion of Rab7 was achieved at P7 (*Figure 3—figure supplement 1A*), sympathetic neurons are less dependent on NGF signaling for survival at this postnatal period. Ablation of Rab7 postnatally, during a period of NGF-dependent synaptogenesis, resulted in a dramatic loss of synaptic structures in sympathetic ganglia, showing that Rab7 is required for transmitting retrograde NGF synaptogenic signals. It is possible that a neuronal survival defect occurring between P7 and P14 contributes to the apparent loss of synaptic markers, or the extent of their colocalization, in the in vivo experiments. Future work will be needed to more rigorously test whether there is a survival-independent role of Rab7 during the establishment of sympathetic neuron connectivity. Nevertheless, a role for Rab7 during retrograde TrkA transport, survival and synapse formation provides additional support for the notion that MVBs are major carriers of retrograde NGF signals. Molecularly, one mechanism by which Rab7 mediates axonal transport of TrkA endosome is through its effector protein RILP, which also binds to dynein. In at least one reported case, the Rab7-RILP-dynein interaction facilitates movement of late endosomes on microtubules and maintains their perinuclear localization (*Jordens et al., 2001*; *Johansson et al., 2007*). Consistent with this, we found that RILP is recruited to TrkA$^+$ MVBs in a Rab7-dependent manner and is required for retrograde TrkA transport. Therefore, we suggest that Rab7 controls maturation and transport competency of TrkA$^+$ MVBs through recruitment of RILP.

## Multivesicular bodies as sorting and signaling platforms in neuronal soma

A long-standing view is that MVBs either repress signaling via receptor downregulation or, in some cases, facilitate signaling by degrading negative regulators (*Taelman et al., 2010*; *Katzmann et al., 2002*). Our findings provide evidence that MVBs can promote growth factor signal transduction directly. A signaling role for MVBs is supported by the presence of P-Trk phospho-tyrosine residues and P-PLCγ associated with retrograde TrkA$^+$ MVBs in cell bodies. Moreover, our pulse-block EM experiments revealed that MVBs give rise to, either directly or indirectly, a new type of endosome that has a single membrane structure that does not associate with Rab5 and that exhibits a topology of TrkA receptors in which the kinase domain is oriented toward the cytoplasm and thus poised for RTK signaling within the cytoplasm. MVBs can fuse with the plasma membrane and release

intraluminal vesicles as exosomes. Therefore, theoretically, the TrkA⁺ SVs we observed within cell bodies could be formed via exosome fusion with the cell body plasma membrane and subsequent receptor internalization. However, both our immunocytochemical and APEX2 EM findings indicate a lack of Rab5⁺ Flag-TrkA endosomes in cell bodies within the time frame of our measurements (*Figure 6* and *Figure 6—figure supplement 1*). Therefore, it is unlikely that the Flag-TrkA⁺ single vesicle population we observed in the pulse-block EM experiments is derived from an exosome route.

To our knowledge, an MVB-to-SV maturation step has been observed in only two other examples. First, in naive dendritic cells, MHC II receptor proteins are stored within intraluminal compartments of MVBs. Upon stimulation, these receptors can translocate from MVBs to the plasma membrane via an endosome intermediate (*Kleijmeer et al., 2001*; *Turley et al., 2000*). In a second case, Ebola virus, upon cellular entry, becomes enriched in Rab7⁺ late endosomes and this is an obligatory step for infection (*Saeed et al., 2010*; *Spence et al., 2016*). The virus can later exit late endosomes and re-enter the cytoplasm. The virus is not sorted into ILVs, but its envelope, a single-pass membrane structure, is thought to be fused with the MVB limiting membrane through Niemann–Pick disease, type C proteins NPC I (*Carette et al., 2011*). These studies, together with the present work, indicate that MVBs are dynamic, versatile structures that, depending on the nature of their cargoes, function in cargo transport, sorting, degradation, and signaling. Moreover, we suggest that TrkA receptors localized to both SVs and the limiting membrane of MVBs provides a potential platforms for recruitment and activation of downstream effector proteins following retrograde the period of TrkA⁺ MVB transport to cell bodies. Determining the mechanisms by which TrkA⁺ MVBs give rise to SVs within the cell body, and defining the relative contributions of TrkA receptors on MVB limiting membranes and SV membranes for cell body TrkA signaling will be important future directions of this work.

## TrkA signaling directs the maturation and fate of retrogradely transported MVBs

While the existence of signaling endosomes following receptor endocytosis is well documented, mechanisms that underlie their maturation and metabolism have remained unclear, as has the significance of receptor activity and signaling from endosomal platforms in general (*Sorkin and Von Zastrow, 2002*). This is at least partly due to the close spatial and temporal proximity between endosomal signaling events and signals emanating from the plasma membrane, which are difficult to distinguish in cells with a simple morphology. Studies of target-derived growth factor signaling in neurons provide a unique opportunity to address mechanisms of endosome maturation, metabolism and signaling because the plasma membrane of distal axons is physically well separated from retrogradely transported endosomes that reside within the soma. Our pulse-block paradigm, which takes advantage of the separation between the site of plasma membrane signaling and endosome formation (distal axons) and a distant site of endosome signaling, maturation and degradation (cell bodies), allowed us to monitor stages of TrkA⁺ endosome maturation and degradation in cell bodies in a synchronized fashion. Our findings that TrkA⁺ SVs emerge from MVBs within cell bodies and that delayed lysosomal sorting occurs for retrograde TrkA⁺ MVBs, but not transferrin-, BSA-, or CTB-containing MVBs, suggest that the emergence of SVs from MVBs is under the control of some unique feature of TrkA⁺ MVBs. Indeed, we found that inhibition of TrkA kinase activity in cell bodies using a selective chemical genetic approach caused TrkA⁺ MVBs to behave similarly to transferrin⁺, BSA⁺, and CTB⁺ MVBs, which do not give rise to SVs but instead rapidly sort to lysosomes. Moreover, cell-body-generated NGF signals partially rescued the rapid lysosomal fate observed for signaling-deficient TrkA⁺ MVBs while TrkA signals emanating from the cell body had no effect on the behavior of MVBs carrying non-TrkA cargoes, suggesting that NGF/TrkA signaling can function both in cis and in trans to influence maturation of retrograde TrkA MVBs. Therefore, active TrkA signaling is required for the emergence of TrkA⁺ SVs from MVBs and the prevention of precocious lysosomal sorting, both of which, we suggest, promote further TrkA signaling.

One effector protein preferentially associated with TrkA⁺ endosomes in cell bodies, coronin, has been implicated in delaying lysosomal degradation of TrkA and promoting survival of sympathetic neurons (*Suo et al., 2014*). We propose a TrkA signaling-dependent, endosome-autonomous, feed-forward mechanism that augments and sustains retrograde NGF signal transduction within cell bodies, through TrkA⁺ MVB and SV recruitment of coronin and possibly other TrkA effectors. Taken together, our findings reveal that MVBs mediate long-range axonal transport of active TrkA, MVBs function in the cell soma as signaling and sorting platforms that give rise to signaling competent

SVs, and the nature of MVB cargoes dictates MVB function and fate. It is noteworthy that defects in endosome function in neuronal processes and soma have been implicated in neurodegenerative diseases and disorders with neurological manifestations (*Cosker and Segal, 2014*). Moreover, dysfunction of early endosome activity has been implicated in several types of neurological disorders (*Cooper et al., 2001*; *Salehi et al., 2006*; *Israel et al., 2012*). Our findings of a central role of MVBs in retrograde neurotrophin signaling suggest a need to consider potential contributions of dysfunctional MVB biogenesis, trafficking, sorting, and signaling to neurological diseases.

# Materials and methods

## Key resources table

| Reagent type (species) or resource | Designation | Source or reference | Identifiers | Additional information |
|---|---|---|---|---|
| Genetic reagent (*Mus musculus*) | Ntrk1*Flag* | (*Sharma et al., 2010*) PMID: 20696380 | | Mice were handled and housed in accordance with Harvard Medical School and Johns Hopkins University IACUC guidelines |
| Genetic reagent (*M. musculus*) | Th*2a-CreER* | (*Abraira et al., 2017*) PMID: 28041852 | | |
| Genetic reagent (*M. musculus*) | Rab7*flox* | (*Roy et al., 2013*) PMID: 23615463 | | |
| Antibody | NeuN | Abcam (Cambridge, MA) | | 1:1000 |
| Antibody | Rab5 | | | 1:500 |
| Antibody | Rab7 | | | 1:200 for ICC; 1:1000 for immunoblot |
| Antibody | Lamp1 | | | 1:1000 |
| Antibody | P-Trk Y490 | Cell signaling (Danvers, MA) | | 1:2000; 1:100 for immunoEM |
| Antibody | P-Trk Y785 | | | |
| Antibody | P-PLCγ | | | 1:500; 1:50 for immunoEM |
| Antibody | VAChT | Enzo Life Sciences (Farmingdale, NY) | | 1:1000 |
| Antibody | Homer1 | Synaptic Systems (Germany) | | 1:500 |
| Antibody | Flag | Sigma (St. Louis, MO) | | 1 ug/ml |
| Antibody | Alexa conjugated secondary antibodies (488, 555, 647) | Thermo Fisher Scientific (Waltham, MA) | | 1:1000 |
| Antibody | IgG F(ab')2–6 nm/10 nm gold secondary antibodies | Aurion (Netherlands) | | 1:50 |
| Antibody | Protein A-5nm/10 nm gold | Made by The Harvard Medical School EM Facility | | 1:50 |
| Recombinant DNA reagent | FUW | (*Lois et al., 2002*) PMID: 11786607 | | Addgene 14882 |
| Recombinant DNA reagent | APEX2 | (*Lam et al., 2015*) PMID: 25419960 | | Addgene 49385 |
| Recombinant DNA reagent | FUW-EGFP-Rab5 | This paper | | |
| Recombinant DNA reagent | FUW-EGFP-Rab7 | This paper | | |
| Recombinant DNA reagent | FUW-CD63-EGFP | This paper | | |
| Recombinant DNA reagent | FUW-CD63-mCherry | This paper | | |
| Recombinant DNA reagent | FUW-RILP-EGFP | This paper | | |

*Continued on next page*

*Continued*

| Reagent type (species) or resource | Designation | Source or reference | Identifiers | Additional information |
|---|---|---|---|---|
| Recombinant DNA reagent | FUW-APEX2-Rab5 | This paper | | |
| Recombinant DNA reagent | FUW-APEX2-Vps35 | This paper | | |
| Recombinant DNA reagent | FUW-Flag-TrkB/A-WT | This paper | | The TrkB/A chimeric receptor comprises the extracellular domain of TrkB (nucleotide 1–1242) and the transmembrane and intracellular domains of TrkA (nucleotide 1156–2400) |
| Recombinant DNA reagent | FUW-Flag-TrkB/A-F592A | This paper | | |
| Sequence-based reagent | Primers for genotyping $Th^{2a-CreER}$ | | | CATGCCCATATCCAATCTCC and CTGGAGCGCATGCAGTAGTA |
| Sequence-based reagent | Primers for genotyping $Rab7^{flox}$ | | | CTCACTCACTCCTAAATGG and TTAGGCTGTATGTATGTGC |
| Sequence-based reagent | shRNAs for Rab7 | | | GAAGTTCAGTAACCAGTACAA; GCGGCAGTATTCTGTACAGTA; GCCCTTAAACAGGAAACAGAA; TGAACCCATCAAACTGGACAA; TGCTGTGTTCTGGTGTTTGAT |
| Sequence-based reagent | shRNAs for RILP | | | CAGCTATGCAGGAGGCTTAAC; AGATCAAGGCCAAGATGTTAG; CCAGAATTTCTTTGGCTTATG; TTCAGCAGGGAAGAGCTTAAG; AGGAGCGGAATGAGCTCAAAG |
| Sequence-based reagent | Scrambled shRNA | | | CCTAAGGTTAAGTCGCCCTCG |
| Chemical compound, drug | K252a | EMD Millipore (Billerica, MA) | | |
| Chemical compound, drug | 1NMPP1 | | | |
| Chemical compound, drug | Nocodazole | | | |
| Chemical compound, drug | Saponin | MP Biomedicals (Santa Ana, CA) | | |
| Chemical compound, drug | 3.3'-Diaminobenzidine (DAB) | | | |
| Chemical compound, drug | 6 nm gold-conjugated transferrin, CTB, BSA | Electron Microscopy Sciences (Hatfield, PA) | | |

## Molecular cloning, transfection and lentiviral infection

cDNAs of mouse Rab5, Rab7, CD63 and Hrs and human Rab7 were purchased from GE Healthcare. A plasmid containing the APEX2 cDNA was a gift from Alice Ting (Addgene plasmid # 49385). Individual cDNAs or fusion transgenes were cloned into the lentiviral vector FUW by In-Fusion cloning kit (Takara). Sequences of 21mer siRNAs against mouse Rab7 or RILP were obtained from TRC library (*Moffat et al., 2006*). shRNA oligos were synthesized, annealed and cloned into the lentiviral pLLX vector as described (*Zhou et al., 2006*).

DNA plasmids were transfected into neuronal cells on DIV three using Lipofectamine 2000 (1 μg DNA: 4 μl Lipofectamine per well).

Lentivirus was generated, harvested and concentrated as previously described (*Salmon and Trono, 2007*). Neuronal cultures were infected on DIV three and experiments were performed 48 ~ 72 hr later.

## Immunohistochemistry

Superior cervical ganglia (SCG) were dissected, fixed in 4% PFA at room temperature for 1 hr, and cryoprotected in 30% sucrose at 4°C overnight. Individual ganglion was embedded in OCT (Tissue Tek) and sectioned at 10 μm. Tissue sections were rehydrated in PBS, blocked 5% serum and 0.05%

TritonX-100 for 1 hr and incubated with primary antibodies at 4°C overnight. The next day, sections were washed and incubated with fluorescent secondary antibodies for 1 hr at room temperature. After extensive wash, sections were mounted. Images were acquired by laser scanning confocal microscopy (Zeiss LSM 700).

## Neuronal cell culture

Sympathetic neurons were cultured as described previously (*Sharma et al., 2010*; *Harrington et al., 2011*). Briefly, SCGs harvested from P0-P4 mice were dissociated and plated in mass cultures or compartmentalized microfluidic devices. Neuronal cultures were maintained in DMEM supplemented with 10% FBS and NGF (50 ng/ml). Cytosine β-D-arabinofuranoside (AraC) was added from DIV 1–3 to eliminate proliferating fibroblast and glia cells. DRG sensory neurons were cultured as previously described (*Huang et al., 2015*).

## The Flag-TrkA endosome transport assay

The Flag-TrkA assay was conducted as follows: an anti-Flag antibody was applied to the distal axon compartment (DA) of the microfluidic chamber and cells were incubated for 45 min at 4°C. Axons were washed extensively to remove unbound antibody and NGF was applied to the DA side. Cells were then incubated at 37°C for indicated times to allow Flag-TrkA internalization, maturation and trafficking. To visualize internalized Flag-TrkA in distal axons, the DA compartment was washed with 0.5M NaCl/0.2M acetic acid to remove surface bound antibody prior to fixation.

## Immunocytochemistry

To visualize proteins associated endosomes, a 0.025% saponin wash was performed first for 2 min. Cells were then fixed with 4% PFA for 10 min, blocked in 0.05% saponin and 2% serum, and subjected to antibody staining. TrkA$^+$ neurons in DRG cultures were identified by TrkA immunostaining and only those neurons were counted for quantification. 16 bit Images were acquired at resolution 1024*1024 by laser scanning confocal microscopy (Zeiss LSM 700).

## Live cell imaging

To track Flag-TrkA endosome movement in real time, an anti-Flag antibody pre-conjugated to Alexa Fluoro secondary antibody was applied to the distal axon compartment during the 4°C incubation step of the Flag-TrkA assay. Prior to imaging, culture medium was replaced by artificial cerebrospinal fluid (ACSF) that is phenol red free and $CO_2$ independent. Axons and cell bodies were imaged by spinning disk microscopy (Zeiss) at 37°C in an environmental chamber at two frames per second (512 × 512 pixels) using a 63X oil immersion objective (1.40 NA). Time-lapse images were imported to ImageJ (NIH) and individual vesicles were tracked manually and analyzed by the MtrackJ plugin (*Meijering et al., 2012*).

## Electron microscopy and APEX2 detection

To detect retrogradely transported Flag-TrkA, transferrin, CTB or BSA, an anti-Flag antibody pre-conjugated to Protein A-5 nm gold secondary antibody, transferrin-gold (6 nm), CTB-gold (6 nm) or BSA-gold (6 nm) was applied to distal axons during the 4°C incubation step of the Flag-TrkA assay, respectively. After the transport assay, cells were fixed in 2.5% glutaraldehyde in 0.1M cacodylate buffer for 30 min at room temperature. Cells were then post-fixed with 1% osmium tetroxide, stained with 2% uranyl acetate and 2% tannic acid to enhance membrane contrast, dehydrated in series of ethanol followed by embedding in EPON resin. The next day, coverslips were removed and areas containing cells were randomly selected and mounted. Ultrathin sections (70 nm) were collected, stained with lead citrate, and were examined on a JOEL electron microscope.

To detect protein localization by APEX2 reaction, diaminobenzidine (DAB) staining was performed as described previously (*Lam et al., 2015*): after fixation, cells were incubated in a Tris buffer containing DAB (0.7 mg/ml) and $H_2O_2$ (0.7 mg/ml) for 15–40 min until the development of brown reaction product. Cells were then washed extensively to prevent further reaction. Samples were subjected to EM procedures as above. Staining with uranyl acetate, tannic acid and lead citrate was omitted to maximize EM contrast generated by DAB staining.

## Immunoelectron microscopy

Flag-TrkA assay was performed as above. Cells were fixed in 4% PFA and 0.15% glutaraldehyde for 30 min. Cells were then washed with 0.1% sodium borohydride for 30 min at room temperature to quench excessive free aldehyde and blocked in PBS with 10% serum, 0.05% saponin and 0.5% gelatin for 2 hr. Cells were incubated in primary antibodies in blocking solution overnight. After extensive washing, cells were incubated with 10 nm gold secondary antibodies overnight and washed again. Cells were then processed for EM.

## The pulse-block assay

The Flag-TrkA assay was performed as above. Nocodazole (10 μM) was added to distal axons at indicated time points post-NGF application. Cells were fixed at indicated time points and processed for ICC, EM or immunogold labeling as described in previous sections.

## The pulse-block kinase assay

Compartmentalized sympathetic neurons were infected with lentivirus expressing either Flag-TrkB/A-F592A or Flag-TrkB/A-WT. The Flag-TrkA assay was performed as described with the following modification: cells were stimulated with BDNF instead of NGF, and either DMSO or the TrkA F592A kinase activity inhibitor 1NMPP1 (500 nM) was applied to the cell body compartment at varying time points as indicated during the course of the experiment. The cells were then subjected to ICC or TEM as above.

## Image analysis

For all quantifications, axon segments or cell body areas were randomly selected. Images were first background subtracted by the Rolling Ball method (*Sternberg, 1983*) and then smoothed before further analyses in ImageJ. Colocalization was assessed by Pearson's correlation.

## Acknowledgements

We thank David Paul, Thomas Lloyd, William Mobley and members of the Ginty laboratory for valuable scientific discussions and comments on this manuscript. We thank Paul Barnes for providing *Rab7*$^{f/f}$ mice, Amanda Zimmerman for *Th*$^{2a-CreER}$ mice, Huaqiang Fang, Richard Huganir and Daniel Tom for providing advice and access to the spinning disk microscope, Maria Ericsson at the HMS Electron Microscopy Facility for technical support with ultrastructural analyses, and Alex Kolodkin, Martin Riccomagno and Sjaak Neefjes for sharing reagents. This work was supported by NIH grant NS97344 (DDG) and the Edward R and Anne G Lefler Center for Neurodegenerative Disorders (DDG). DDG is an investigator of the Howard Hughes Medical Institute.

## Additional information

### Competing interests

David D Ginty: Reviewing editor, *eLife*. The other authors declare that no competing interests exist.

### Funding

| Funder | Grant reference number | Author |
| --- | --- | --- |
| Howard Hughes Medical Institute | | David D Ginty |
| National Institute of Neurological Disorders and Stroke | NS97344 | David D Ginty |
| Edward R and Anne G Lefler Center for Neurodegenerative Disorders | | David D Ginty |

The funders had no role in study design, data collection and interpretation, or the decision to submit the work for publication.

## Author contributions
Mengchen Ye, Conceptualization, Data curation, Formal analysis, Validation, Investigation, Visualization, Methodology, Writing—original draft, Writing—review and editing; Kathryn M Lehigh, Methodology; David D Ginty, Conceptualization, Resources, Funding acquisition, Writing—original draft, Writing—review and editing

## Author ORCIDs
Mengchen Ye [iD] http://orcid.org/0000-0002-2820-1192
David D Ginty [iD] http://orcid.org/0000-0001-9723-8530

## Ethics
Animal experimentation: Mice were handled and housed in accordance with Harvard Medical School IACUC guidelines and described in protocol number 05041.

## Decision letter and Author response
Decision letter https://doi.org/10.7554/eLife.33012.022
Author response https://doi.org/10.7554/eLife.33012.023

# Additional files

## Supplementary files
• Transparent reporting form
DOI: https://doi.org/10.7554/eLife.33012.020

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
