## [Decision Letter]

Thank you for submitting your article "Multivesicular bodies mediate long-range retrograde NGF-TrkA signaling" for consideration by *eLife*. Your article has been reviewed by three peer reviewers, and the evaluation has been overseen by a Reviewing Editor and Gary Westbrook as the Senior Editor. The following individual involved in review of your submission has agreed to reveal their identity: Lee Francis (Reviewer #2). The reviewers have discussed the reviews with one another and the Reviewing Editor has drafted this decision to help you prepare a revised submission.

Summary:

This interesting manuscript from Ye et al. examines the role of multivesicular bodies (MBVs) in long-range NGF/TrkA signaling. The manuscript details studies in which TrkA receptor signaling endosomes are a sub-type of multivesicular bodies, which are retrogradely transported, evade lysosomal fusion and evolve into TrkA-positive single-membrane vesicles that are capable of signaling. The cell biological and EM studies are clearly designed and the results make a significant contribution to our understanding of TrkA retrograde trafficking as it had been previously thought that Rab5^+^ early endosomes mediate the retrograde propagation of TrkA signaling.

Essential revisions:

The reviewers cited two major concerns:

First, additional in vivo validation is needed to strengthen the conclusions. There is an EM showing a MVB in gold labeled with Flag antibody in the axon. It is critical to show several more representative images, more feeding time points 30' – 3 hours) and different regions of the axon (distal, intermediate and proximal).

Second, further details are needed to explain how the TrkA receptors in MVB's evolve into SV's in the cell bodies. It has been previously shown that retrogradely transported TrkA receptors fused to the plasma membrane in the cell body. It would be interesting to assess whether the TrkA in MVBs route to the plasma membrane prior to SVs. In addition, the abundance of the TrkA receptors in the intralumenal vesicles in relation to the endosomal limiting membrane (Figure 1 and higher mag panel) suggests that fusion of the MVB at the cell surface would result in the extracellular release of the majority of the MVB bound TrkA receptors. If so, the formation of SVs labeled with TrKA would require fusion of the exosome back to the cell surface and subsequent internalization of the receptor into the cell body. At the very least, these issues of topology and recycling and molecular diversification/maturation should be discussed to add more impact to the impact of the study.

Other revisions and statistical comments:

In addition to the major issues, several experimental points need attention:

1) Nocodazole treatment is an interesting approach but more controls are necessary. Does nocodazole effect TrkA movement in the soma or dendrites? Nocodazole may diffuse once inside the neuron. As alternative strategy, ciliobrevin can be used to stop dynein-mediated transport without the drawback of a widespread disruption of the MT cytoskeleton.

2) Previous work indicated the presence of retrogradely-transported Trks in membrane-containing organelles, such as autophagosomes (Kononenko et al., 2017, Nat Comm) and MBVs (Wang et al., 2016, Nat Comm). Please clarify the origin of TrkA^+^ MBVs and their molecular identity in light of the presence of Rab7 and CD63 in both types of organelles. How can the data be reconciled with those obtained in cortical and hippocampal neurons, where size and proportion of retrogradely transported MBVs versus SVs, seem to be very different (Wang et al., 2016)? Likewise, discrepancies with the work of Claude et al., 1982 (e.g. density of cargoes in different compartments). Informative work on the trafficking of other Trk receptors (Terenzio et al. 2014 EMBO J; Terenzio et al., 2017 Neuron) may also contribute to the Discussion.

3) The contribution of Rab7-negative carriers (25%) to the retrograde transport and signalling is not discussed. Also in the kymograph of Figure 2 half of the moving carriers are Rab7-negative.

4) The effects of RILP KD on transport should be controlled by rescue with siRNA-resistant RILP and/or by expression of DN RILP fragments (e.g. RILP-C33).

5) The identification of lysosomes and MBVs is based on single-plane EM and/or the presence of markers (Rab7, CD63, Lamp1), with a broader distribution. Additional identification of these organelles is therefore recommended (i.e. degradative potential, acidic pH and presence of cathepsins).

6) Since some manipulations (e.g. Rab7 KD or blocking signalling) are severe, it would be crucial to include controls showing that under the conditions blocking TrkA^+^ carrier movement, the axonal transport of other cargoes, e.g. mitochondria, remains intact.

7) Is there any effect in survival of sympathetic neurons depleted of Rab7 between P7 and P14? Can we explain the reduced number of pTrk and the decrease in synaptic markers at least partially by a reduction in survival?

8) What is the identity of the acidic compartments shown in the cell body? Then, it is not clear whether the cargoes present in those degradative organelles are generated from single-membrane vesicles rapidly sorted to lysosomal pathway or from acidification of the MVB/autophagosomes due to fusion with lysosomes. Since Tf appears to be rapidly segregated to different carriers than TrkB in distal axons, and given the fact that during the first hour ~50% of Tf is in MVBs, are different classes of MBVs also retrogradely transported?

Figure 2: what is the length of the feed here? 3 hours? Everything looks overexpressed and cytosolic in contrast to the immunostain. D: I would like to break Flag-Trk/rab co-transport out into anterograde versus retrograde. E: is the colocalization of flag and the gfp-rab correct? How was stationary versus movement judged (add to Materials and methods)? From the kymograph of rab5, most look stationary to me.

Figure 3. No necessity to use a magenta color mask. Please use grayscale to enhance contrast. 3I. Is there a reason to plot the number of cells with puncta, instead of the amount of punctate signal per compartmentalised cell?

Figure 4. Please use organelle instead of endosome to label the Y axis.

Figure 5. "pulse" instead of "pule".

Figure 6. The legend states that the n = 3, but the dot plot shows 6 points per treatment.

Figure supplements. Please review the statistical tests performed. In every legend it is stated that a 'Tukey's post-hoc correction' was performed. Tukey's is actually a post-hoc test. Please amend.

Figure 1—figure supplement 1:

D) In order to get at the maturation of the SE, I'd like to see a more detailed time course on both the distal and proximal axon sides for all markers. Low overall rab11 staining is surprising given the high axonal levels observed in Acsano et al., 2009 (Figure 3). How do you reconcile this?

F) Viruses in general and certainly overexpression of Rabs are known to disrupt membrane trafficking. It is important to compare Flag trafficking in uninfected versus infected controls quantifying retrograde, anterograde, and stationary endosomes. Additionally, throughout the paper virally overexpressed proteins seem blown out making the colocalization work not as convincing. A discussion of thresholding approaches and how overexpression is controlled for is important.

g) 80% Flag positive MVBs are reported in Figure 1 with EM is inconsistent with light microscopy -50% cd63^+^MVBs (also in E). Are not all cd63^+^ carries MVBs? Please comment.

Figure 3—figure supplement 1:

A-E) These are really nice controls. cd63 seems quite overexpressed. In some of these pictures it would be hard not to find co-localization. Is this an expression or a thresholding issue? Doesn't the notion that loss of Rab7 doesn't impact flag co-localization with cd63 undermine the premise that rab7 is targeting SEs to MVBs? I would like to see a video in the rab7 KD.

J-K) The expression levels of RILP seems very cytosolic - could you also provide a video? Would be good to see some quantification for J. The expression/thresholding for these axons look dramatically different between control and shrab7--rilp/TrkA^+^. They also move only.5 micron per second, which is half the speed of what was previously reported (Lehigh et al., 2017).

J-M) Need to show single channels and that are less thresholded.

Figure 4—figure supplement 1. Where phospho-specific antibodies are used, the total protein should be included.

Figure 6—figure supplement 1:

Labels are wrong for how they correlate to text.

A) Is there literature on the movement of Tfn in axons? I thought Tfn receptor was not in axons. If not, then this would not be receptor-mediated endocytosis and transport.

G) Too zoomed into see anything. Either color code, outline, or show more zoomed out image.

Supplementary Figure 7. The model would be very valuable in the main section of the paper and may be used to summarize the findings and guide the Discussion.

Materials and methods. Please include statistical methods in the section, including tests and post-tests, sample sizes and power analysis, software or scripts used, etc.

Materials and methods. There are no details on how the microscopy of fixed samples was performed. Please include at least the resolution, image treatment if any, randomisation of electron microscopy frames, segmentation protocols and deconvolution methods if any. Also include the methods used for analysing co-localisation and the appropriate controls of specificity.h. Controls for long-term (8h) fluidic isolation of MFCs should be provided (e.g. for 1NMPP1 experiments).

There is a need to be explicit in distinguishing the proportion of the receptor versus the proportions of carriers that corresponds to each organelle. Is not the same to say that "most of TrkA is in MVBs" and that "most of the retrograde carriers of this receptor are MVBs". Also the criteria for selecting the axon segments and image processing (e.g. denoising, background subtraction, deconvolution) should be indicated. When colocalisation quantification is performed, please indicate the methods used.

Please review the pertinence of the statistical tests used for each dataset to comply *eLife* policies (e.g. Figure 3 populations are compared, hence Student's t-test should be used; Figure 4 and Figure 5: since there are multiple datasets in these plots, please use Student's t-test for each pair of comparisons; Figure 5. please indicate the number of replicates for each data point and justify your conclusions. For statistical significance, please use 2-way ANOVA; Figure 7: two factors account for the variation in this dataset: treatment and receptor. Please use 2-way ANOVA; Figure 7: three factors accounts for the variation: receptor, treatment and time. Please use 3-way ANOVA). No sample-size estimation, replicates definition, justification for statistical analysis methods or data description have been provided.

---

## [Author Response]

Essential revisions:The reviewers cited two major concerns:First, additional in vivo validation is needed to strengthen the conclusions. There is an EM showing a MVB in gold labeled with Flag antibody in the axon. It is critical to show several more representative images, more feeding time points 30' – 3 hours) and different regions of the axon (distal, intermediate and proximal).

We agree with the reviewers that the observation of retrograde Flag-TrkA MVBs in vivo would strengthen the conclusions of the study. To address this, we attempted injection of anti-Flag antibody to back hairy skin of *TrkA^FLAG^* mice, however we were unable to detect a Flag signal in axons or cell bodies of DRG sensory neurons by immunohistochemistry and thus we did not attempt to visualize Flag-TrkA by EM in vivo. A systematic approach testing different labeling methods such as quantum dot-NGF is likely to be necessary to achieve this goal. This will be a focus of future studies.

Regarding in vitro visualization of Flag-TrkA endosomes in different compartments of neurons over time, we have included more representative images in Figure 1—figure supplement 1. After antibody feeding and at the onset of NGF application, gold-labeled Flag antibody is present only on the plasma membrane of distal axons. After five minutes, gold particles were found in single-membrane vesicles in the vicinity of the plasma membrane, most likely early endosomes. At 30 minutes post NGF application, gold particles were found in both early endosomes and MVBs in distal axons but not proximal axons or cell bodies (Figure 1—figure supplement 1). At the 1 hr time point, the majority of gold particles in distal axons were found in MVBs. Importantly, only Flag-TrkA MVBs were observed in proximal axons and cell bodies in the cell body compartment (Figure 1—figure supplement 1). At the 3 hr time point, the majority of gold particles in distal axons and proximal axons were found in MVBs (Figure 1—figure supplement 1). A mixture of Flag-TrkA single-membrane vesicles and MVBs were found in cell bodies, as seen in Figure 5. It should be noted that the quality of EM images for axons is often lower than that of cell bodies because of the presence of thick, electron-dense microtubule tracks which often obscure visualization of endosomal structures and gold particles.

We have reported these findings in the last paragraph of the subsection “Ultrastructural analysis of retrograde Flag-TrkA endosomes” and the data are shown in Figure 1—figure supplement 1 of the revised manuscript.

Second, further details are needed to explain how the TrkA receptors in MVB's evolve into SV's in the cell bodies. It has been previously shown that retrogradely transported TrkA receptors fused to the plasma membrane in the cell body. It would be interesting to assess whether the TrkA in MVBs route to the plasma membrane prior to SVs. In addition, the abundance of the TrkA receptors in the intralumenal vesicles in relation to the endosomal limiting membrane (Figure 1 and higher mag panel) suggests that fusion of the MVB at the cell surface would result in the extracellular release of the majority of the MVB bound TrkA receptors. If so, the formation of SVs labeled with TrKA would require fusion of the exosome back to the cell surface and subsequent internalization of the receptor into the cell body. At the very least, these issues of topology and recycling and molecular diversification/maturation should be discussed to add more impact to the impact of the study.

In axons, we observed a small percentage of Flag-TrkA punctae (~8%) associated with Rab11, a marker for recycling endosomes (Figure 2—figure supplement 1), suggesting that at least some of retrogradely transported TrkA receptors can be recycled back to plasma membrane.

As for retrogradely transported TrkA MVBs in cell bodies, our pulse block EM analysis showed that very few, if any, Flag-TrkA receptors in cell bodies are associated with single-membrane vesicles, which include early endosomes and recycling endosomes, or present on the plasma membrane before the 3 hr time point (Figure 5). Furthermore, we never detected Flag-TrkA MVBs in close proximity to the plasma membrane by EM. Complementary immunocytochemical experiments using the same pulse-block paradigm under non-permeabilizing condition also showed no Flag signal on the plasma membrane at the 1 or 2 hr time points (data not shown). Thus, although we could not formally exclude the possibility of delivery of retrograde TrkA onto plasma membrane via MVB fusion, our data provide no evidence for the existence of such a process.

Between the 3 hr and 8 hr time points in our pulse-block EM analyses, the emergence of TrkA single vesicles could, theoretically, be formed via a route consisting MVB fusion with plasma membrane and exosome release, exosome fusion with plasma membrane and subsequent receptor internalization. If that were the case, internalized TrkA should follow the canonical endocytic pathway by sorting into Rab5^+^ early endosomes. However, both our immunocytochemical and APEX2 EM results indicate a lack of Rab5^+^ Flag-TrkA endosomes during this time period (Figure 6 and Figure 6—figure supplement 1). Therefore, it is highly unlikely that the Flag-TrkA single vesicle population we observed in the pulse-block EM experiments is derived from exosomes.

Lastly, at the 3 hr time point in the pulse-block immunocytochemical experiments, ~13% of Flag-TrkA were Rab11^+,^ suggesting their potential recycling route to the plasma membrane (Figure 6). This percentage is similar to previously published results using the same Flag-TrkA endosome assay without the pulse-block treatment (Suo et al., 2014). Therefore, our findings suggest that a subset of retrogradely transported TrkA receptors may be recycled and potentially re-inserted to the plasma membrane, but this process is likely to be mediated by TrkA single vesicles formed de novo in cell bodies, not by MVB fusion with plasma membrane.

As requested, we have added a new paragraph addressing this point to the Discussion section.

Other revisions and statistical comments:In addition to the major issues, several experimental points need attention:1) Nocodazole treatment is an interesting approach but more controls are necessary. Does nocodazole effect TrkA movement in the soma or dendrites? Nocodazole may diffuse once inside the neuron. As alternative strategy, ciliobrevin can be used to stop dynein-mediated transport without the drawback of a widespread disruption of the MT cytoskeleton.

We acknowledge the concerns raised by the reviewers regarding the use of nocodazole. As seen in supplementary Figure 5, nocodazole treatment of the cell body compartment completely blocked retrograde Flag-TrkA movement in proximal axons and cell bodies as well as endocytic sorting of TrkA to lysosomes when anti-Flag antibody was directly applied to cell bodies. Importantly, nocodazole treatment of distal axons did not affect microtubule-dependent endosome formation and trafficking within the cell body compartment, indicating effective compartmentalization of transport blockade (See Figure 5—figure supplement 1). The sympathetic neurons in these experiments have few if any dendrites because extension of dendrites requires more than 10 days in this culture paradigm. This point is clarified in the second paragraph of the subsection “Retrogradely transported TrkA^+^ endosomes in cell bodies evolve from MVBs into simple, single membrane vesicle structures and evade lysosomal sorting”.

We also attempted to use ciliobrevin (50 μM) in our initial experimental design as it is a potent inhibitor of dynein (Firestone et al., Nature, 2012). However, in our hands, it took ~ 45 minutes to completely halt retrograde movement of Flag-TrkA endosomes in axons following application of ciliobrevin. Similar kinetics have been observed in its application in cell cycle arrest and inhibition of axonal transport in sensory neurons (Firestone et al., Nature, 2012; Sainath et al., Dev. Neurobio., 2014). This time frame gave more than a “pulse” of retrograde Flag-TrkA and confounded our EM time-course analyses because of the presence of continuously incoming TrkA MVBs within cell bodies after the initial time point. In contrast, retrograde transport was fully blocked after less than 15 minutes of nocodazole application.

2) Previous work indicated the presence of retrogradely-transported Trks in membrane-containing organelles, such as autophagosomes (Kononenko et al., 2017, Nat Comm) and MBVs (Wang et al., 2016, Nat Comm). Please clarify the origin of TrkA^+^ MBVs and their molecular identity in light of the presence of Rab7 and CD63 in both types of organelles. How can the data be reconciled with those obtained in cortical and hippocampal neurons, where size and proportion of retrogradely transported MBVs versus SVs, seem to be very different (Wang et al., 2016)? Likewise, discrepancies with the work of Claude et al., 1982 (e.g. density of cargoes in different compartments). Informative work on the trafficking of other Trk receptors (Terenzio et al. 2014 EMBO J; Terenzio et al., 2017 Neuron) may also contribute to the Discussion.

Rab7 and CD63 proteins are known to be associated with MVBs, late endosomes and late endosome/lysosome fusion intermediate structures (Pols MS., Exp. Cell. Res. 2009) and there are discrepancies and inconsistencies in the literature regarding the definition of a particular type of endosome based on its ultrastructure or molecular features. These discrepancies prompted us to evaluate the identity of retrograde TrkA endosomes by both light and electron microscopy using the same in vitro Flag endosome transport assay in sympathetic neurons to minimize technical and biological variation. Our EM analyses showed that nearly all retrogradely transported Flag-TrkA resides within MVBs that are devoid of mature late endosome and lysosome features such as increased electron density and a multi-laminar ultrastructure, nor do they contain irregular-shaped intraluminal structures typically associated with autophagosomes. These observations suggest that Flag-TrkA MVBs are likely to be a subset of Rab7 or CD63^+^ endosomes.

The differences between this study and others addressing TrkB endosomes are most likely due to the very different techniques and neuronal populations used, and potential differences between TrkA and TrkB. In our hands, TrkA trafficking in DRG sensory neurons is similar to what we have observed in sympathetic neurons: MVBs are the major retrograde carrier of TrkA in axons, and Rab7 plays an essential role in retrograde TrkA transport in both cell types in vitro. However, although we favor the idea that technical differences account for differences in our findings and those reported in cortical and hippocampal neurons with a different Trk receptor, the nature of retrograde TrkA endosomes and underlying mechanisms could be different for TrkB endosomes in CNS neuronal populations. This point is now discussed in the Discussion section.

3) The contribution of Rab7-negative carriers (25%) to the retrograde transport and signalling is not discussed. Also in the kymograph of Figure 2 half of the moving carriers are Rab7-negative.

From our co-localization analyses, retrogradely transported TrkA is also associated with early endosomes, recycling endosomes and lysosomes, albeit only very small percentage of endosomes. Knockdown of EEA1, a major protein for early endosome function, abolished retrograde transport but also prevented formation of TrkA MVBs in distal axons (data not shown). We plan to study the roles of each of these types of endocytic compartments for the initiation of retrograde transport and signaling in the future.

Live imaging of Flag-TrkA movement was performed in proximal axon regions where axons often bundle together. Our interpretation is that not every axon expresses the labeled endosomal marker proteins because the infection efficiency is less than 100% and therefore Flag-TrkA in these axons is not always associated with fluorophore-tagged endosomal markers. This point is clarified in the text.

4) The effects of RILP KD on transport should be controlled by rescue with siRNA-resistant RILP and/or by expression of DN RILP fragments (e.g. RILP-C33).

We used a pool of five shRNAs for the knockdown experiments, which minimized potential off-target effects. Also, our immunocytochemical and live imaging results, taken together, indicate that RILP is a key effector downstream of Rab7 that mediate retrograde TrkA movement.

5) The identification of lysosomes and MBVs is based on single-plane EM and/or the presence of markers (Rab7, CD63, Lamp1), with a broader distribution. Additional identification of these organelles is therefore recommended (i.e. degradative potential, acidic pH and presence of cathepsins).

To address this point, we attempted to measure the pH of retrograde CD63^+^ Flag-TrkA endosomes using LysoSensor, as described in (Ouyang et al., Neuron, 2014). However, due to extensive spectrum overlap needed for four-channel imaging (LysoSensor-blue, LysoSensor-yellow, CD63 and Flag-TrkA), we were unable to reliably measure the acidity of TrkA MVBs. We hope to address this issue in future studies using a more advanced fluorophore tagging system to better understand the biochemical and molecular properties of retrograde TrkA endosomes.

6) Since some manipulations (e.g. Rab7 KD or blocking signalling) are severe, it would be crucial to include controls showing that under the conditions blocking TrkA^+^ carrier movement, the axonal transport of other cargoes, e.g. mitochondria, remains intact.

We agree that this is an important point. We found that Rab7 knockdown did not affect mitochondria movement in axons (Figure 3—figure supplement 1). We have added this finding to the last paragraph of the subsection “Rab7 mediates retrograde TrkA transport and signaling in vivo and in vitro” and the data are now shown in Figure 3—figure supplement 1 of the revised manuscript.

7) Is there any effect in survival of sympathetic neurons depleted of Rab7 between P7 and P14? Can we explain the reduced number of pTrk and the decrease in synaptic markers at least partially by a reduction in survival?

We agree with the reviewers that a survival defect may contribute to the apparent loss of synaptic markers and their colocalization in the in vivo experiments. To test whether there is a survival-independent role of Rab7 in the development of sympathetic circuit, we plan to compare *Th^CreER^; Rab7^f/f^; Bax^+/-^* and *Th^CreER^; Rab7^f/f^; Bax^-/-^* animals in the future. We have noted this possibility in the fourth paragraph of the Discussion.

8) What is the identity of the acidic compartments shown in the cell body? Then, it is not clear whether the cargoes present in those degradative organelles are generated from single-membrane vesicles rapidly sorted to lysosomal pathway or from acidification of the MVB/autophagosomes due to fusion with lysosomes. Since Tf appears to be rapidly segregated to different carriers than TrkB in distal axons, and given the fact that during the first hour ~50% of Tf is in MVBs, are different classes of MBVs also retrogradely transported?

As seen in Figure 5—figure supplement 1 and correctly stated by the reviewers, we never observed retrogradely transported transferrin and TrkA localized to the same MVB, despite robust labeling with both types of gold particles. These findings support the idea that transferrin and TrkA segregate to distinct populations of MVBs in distal axons which then undergo retrograde transport and entry to cell bodies.

Figure 2: what is the length of the feed here? 3 hours? Everything looks overexpressed and cytosolic in contrast to the immunostain. D: I would like to break Flag-Trk/rab co-transport out into anterograde versus retrograde. E: is the colocalization of flag and the gfp-rab correct? How was stationary versus movement judged (add to Materials and methods)? From the kymograph of rab5, most look stationary to me.

For this set of live cell imaging experiments, time-lapse images were captured 2 hrs after NGF application in distal axons. The excessive cytosolic signal is due to protein overexpression and was partially removed by saponin treatment before fixation for fixed-cell staining. We did not perform this treatment for live imaging for obvious cell survival issues.

In Figure 2, the quantification is for EGFP-Rab5^+^ or EGFP-Rab7^+^ Flag-TrkA (see corrected Figure 2 in the revised manuscript). A stationary endosome is defined by movement of less than 0.01 μm in each time frame for all frames for the duration of the imaging session.

Figure 3. No necessity to use a magenta color mask. Please use grayscale to enhance contrast. 3I. Is there a reason to plot the number of cells with puncta, instead of the amount of punctate signal per compartmentalised cell?Figure 4. Please use organelle instead of endosome to label the Y axis.

We have used the same color scheme throughout the figures. Therefore, for the sake of consistency and clarity, we have opted to keep the magenta color for P-Trk and Flag as in the original figure.

For simplicity, we quantified the number of cells with puncta. Reanalysis using puncta per cell gave similar results.

We have changed the axis label for Figure 4 to organelle in the revised manuscript.

Figure 5. "pulse" instead of "pule".Figure 6. The legend states that the n = 3, but the dot plot shows 6 points per treatment.Figure supplements. Please review the statistical tests performed. In every legend it is stated that a 'Tukey's post-hoc correction' was performed. Tukey's is actually a post-hoc test. Please amend.

Figure 5 and Figure 6 and the figure supplements have been revised to reflect these changes.

Figure 1—figure supplement 1:D) In order to get at the maturation of the SE, I'd like to see a more detailed time course on both the distal and proximal axon sides for all markers. Low overall rab11 staining is surprising given the high axonal levels observed in Acsano et al., 2009 (Figure 3). How do you reconcile this?F) Viruses in general and certainly overexpression of Rabs are known to disrupt membrane trafficking. It is important to compare Flag trafficking in uninfected versus infected controls quantifying retrograde, anterograde, and stationary endosomes. Additionally, throughout the paper virally overexpressed proteins seem blown out making the colocalization work not as convincing. A discussion of thresholding approaches and how overexpression is controlled for is important.G) 80% Flag positive MVBs are reported in Figure 1 with EM is inconsistent with light microscopy -50% cd63^+^MVBs (also in E). Are not all cd63^+^ carries MVBs? Please comment.

The main message from this set of co-localization analyses is that retrogradely transported TrkA endosomes in proximal axons are associated with MVB markers such as Rab7, CD63 and Hrs. The maturation of TrkA endosomes after receptor internalization is an interesting question that we plan to study in more detail in the future.

The apparent low Rab11 staining was likely due to saponin treatment before fixation, which removes excessive cytosolic staining. We performed the same experiment without this step and while a more prominent Rab11 signal was observed, the staining was less punctate.

We agree with the reviewers that viral overexpression may lead to artifacts when counting co-localization. To minimize variation, we used the same settings during image acquisition and used the same threshold during quantification. Colocalization was assessed by Pearson’s correlation.

CD63 protein is known to be associated with MVBs, late endosomes and late endosome/lysosome fusion intermediate structures (Pols MS., Exp. Cell. Res. 2009). Our EM analyses revealed that almost all retrogradely transported Flag-TrkA resides in MVBs that are devoid of mature late endosome and lysosome features such as increased electron density and multi-laminar ultrastructure, indicating that Flag-TrkA MVBs are likely to be a subset of CD63^+^ endosomes.

Figure 3—figure supplement 1:A-E) These are really nice controls. cd63 seems quite overexpressed. In some of these pictures it would be hard not to find co-localization. Is this an expression or a thresholding issue? Doesn't the notion that loss of Rab7 doesn't impact flag co-localization with cd63 undermine the premise that rab7 is targeting SEs to MVBs? I would like to see a video in the rab7 KD.

There is variation regarding viral overexpression dependent on the batch of virus and neuronal cell preparation. To minimize variation and have fair comparisons across samples, we used the same settings during imaging acquisition and used the same threshold during quantification.

It is a surprising finding that loss of Rab7 does not affect co-localization between Flag-TrkA and CD63. However, as shown in Figure 3—figure supplement 1, our results suggest that Rab7 is not required for sorting of TrkA receptors to MVBs; rather it is essential for retrograde movement of TrkA MVBs via RILP recruitment.

Figure 4—figure supplement 1. Where phospho-specific antibodies are used, the total protein should be included.

We do not have a decent TrkA antibody that is useful for immunocytochemistry. The amount of Flag signal between the samples was therefore used to provide an adequate, albeit imperfect, control.

Figure 6—figure supplement 1:Labels are wrong for how they correlate to text.A) Is there literature on the movement of Tfn in axons? I thought Tfn receptor was not in axons. If not, then this would not be receptor-mediated endocytosis and transport.G) Too zoomed into see anything. Either color code, outline, or show more zoomed out image.

We have corrected Figure 6—figure supplement 1 and the corresponding text in the revised manuscript.

As stated by the reviewers, previous studies suggest that the transferrin receptor is mostly present in dendrites (e.g., West et al., J Neuroscience, 1997). However, sympathetic neurons in our experiments do not have dendrites and we found that the transferrin receptor is present in axons of sympathetic neurons in vitro. As seen in Figure 6—figure supplement 1, we found that newly internalized transferrin, in both distal axons and cell bodies, is associated with the transferrin receptor. Second, we blocked receptor-mediated endocytosis in distal axons through inhibition of dynamin, which is required for ligand-dependent internalization of the transferrin receptor, and assessed whether internalization of transferrin is perturbed in distal axons (Figure 6—figure supplement 1). Compared to the DMSO vehicle control, treatment of Dyngo, a dynamin inhibitor, completely abolished internalization of transferrin, but not BSA, which is internalized by fluid-phase endocytosis (Figure 6—figure supplement 1). These findings indicate the presence of the transferrin receptor in distal axons of sympathetic neurons and that internalization of transferrin in distal axons requires receptor-mediated endocytosis that is dynamin-dependent.

Supplementary Figure 7. The model would be very valuable in the main section of the paper and may be used to summarize the findings and guide the Discussion.

We agree. We have moved the model to Figure 8 of the main section of the revised manuscript.

Materials and methods. Please include statistical methods in the section, including tests and post-tests, sample sizes and power analysis, software or scripts used, etc.Materials and methods. There are no details on how the microscopy of fixed samples was performed. Please include at least the resolution, image treatment if any, randomisation of electron microscopy frames, segmentation protocols and deconvolution methods if any. Also include the methods used for analysing co-localisation and the appropriate controls of specificity.

We have modified the Materials and methods section to provide detailed experimental and analysis procedures.

There is a need to be explicit in distinguishing the proportion of the receptor versus the proportions of carriers that corresponds to each organelle. Is not the same to say that "most of TrkA is in MVBs" and that "most of the retrograde carriers of this receptor are MVBs". Also the criteria for selecting the axon segments and image processing (e.g. denoising, background subtraction, deconvolution) should be indicated. When colocalisation quantification is performed, please indicate the methods used.

We agree with the reviewers and apologize for the confusion. We have modified the main text to clarify this issue.

For all quantification, axon segments or cell body areas were randomly selected. Images were background subtracted first using the Rolling Ball method and smoothed. Colocalization was performed by Pearson’s correlation. Please see the modified Materials and methods section.

Please review the pertinence of the statistical tests used for each dataset to comply eLife policies (e.g. Figure 3 populations are compared, hence Student's t-test should be used; Figure 5: since there are multiple datasets in these plots, please use Student's t-test for each pair of comparisons; Figure 5. please indicate the number of replicates for each data point and justify your conclusions. For statistical significance, please use 2-way ANOVA; Figure 7: two factors account for the variation in this dataset: treatment and receptor. Please use 2-way ANOVA; Figure 7: three factors accounts for the variation: receptor, treatment and time. Please use 3-way ANOVA). No sample-size estimation, replicates definition, justification for statistical analysis methods or data description have been provided.

We have now analyzed these data according to the reviewers’ suggestions and we have modified the figures, legends and methods in the revised manuscript. These changes have not altered any of the interpretations or conclusions of the original manuscript.

We did not perform power analysis or sample-size estimation prior to experiments.